
# Health and Economic Impacts of Ozone Pollution in
# China: a provincial level analysis
Yang Xie [1]; Hancheng Dai [2*]; Yanxu Zhang [3*]; Tatsuya Hanaoka [1]; Toshihiko Masui [1]
1 Center for Social and Environmental Systems Research, National Institute for Environmental
Studies, 16-2 Onogawa, Tsukuba-City, Ibaraki, 305-8506, Japan
2 dai.hancheng@pku.edu.cn, College of Environmental Sciences and Engineering, Peking
University, Beijing 100871, China;
3 John A. Paulson School of Engineering and Applied Sciences, Harvard University, Cambridge,
MA 02138;
* Corresponding author, dai.hancheng@pku.edu.cn, Room 246, Environmental Building, College
of Environmental Sciences and Engineering, Peking University, Beijing 100871, China; TEL/Fax:
12  (+86) 10-6276-4974

* Corresponding author, zhangyanxv@gmail.com, John A. Paulson School of Engineering and
Applied Sciences, Harvard University, Cambridge, MA 02138;
**Abstract**
Many studies have reported associations between ozone pollution and morbidity and
mortality, but few studies focus on the health and economic effects at China's regional level. This
study evaluates the ozone pollution-related health impacts on China's national and provincial
economy and compares them with the impacts from $PM_{2.5}$. We also explore the mitigation potential
across 30 provinces of China. An integrated approach is developed that combines an air pollutant
emission projection model (GAINS), an air quality model (GEOS-Chem), a health model using
the latest exposure-response functions, medical prices and value of statistical life (VSL), and a
general equilibrium model (CGE). Results show that lower income western provinces encounter
severer health impacts and economic burdens due to high natural background levels of ozone



pollution, whereas the impact in southern and central provinces is relatively lower. Without a
control policy, in 2030 China will experience a 4.24 billion USD Gross Domestic Production
(GDP) loss (equivalent to 0.034%), and a 285 billion USD (equivalent to 2.34% of GDP) life loss.
In contrast, with a control policy, the GDP and VSLs loss will be reduced to 3.72 (0.030%) and
242 billion USD (1.99%), respectively. We conclude that health and economic impacts of ozone
pollution are significantly lower than $PM_{2.5}$, but are much more difficult to mitigate. The Chinese
government should promote the air pollution control policies that jointly reduce both PM2.5
pollution and ozone pollution, and the public should adjust their lifestyle according to the air
quality information.
**Keyword**: Ozone pollution; Health impact; Economic impact; GEOS-Chem model; CGE
model

# 1   Introduction

Ozone is a common air pollutant all over the world, including in both developing and
developed countries. Many studies have reported associations between outdoor ozone
concentrations and morbidity and mortality(Cakmak, et al. 2016; Silva, et al. 2013). Ozone
pollution has been associated with a series of health endpoints, respiratory-related hospital
admissions, cardiovascular disease, lost school days, restricted activity days, asthma-related
emergency department visits, and premature mortality(Hubbell, et al. 2005; Orru, et al. 2013;
Rosenthal, et al. 2013; WHO 2013). Ozone exposure is also related to respiratory symptoms and
the use of asthma medication for asthmatic school children using maintenance medication(Gent,
et al. 2003). McDonnell et al.(McDonnell, et al. 1999) also found long-term exposure to ozone
may be associated with the development of asthma in adult males. Berman et al.(Berman, et al.





2012) evaluated the health benefits from large-scale ozone reduction in the U.S. Fann et al(Fann,
et al. 2012) estimated 4,700 ozone-related deaths resulted from 2005 air quality levels and 36,000
life years were lost from ozone exposure in the United States. Fann et al.(Fann and Risley 2013)
estimated that reductions in monitored $PM_{2.5}$ and ozone from 2000 to 2007 were associated with
22,000-60,000 $PM_{2.5}$ and  880-4,100 ozone net avoided premature mortalities in the United States.

Various studies have attempted to quantify the economic impact of air pollution. Selin et

al.(Selin, et al. 2009)assessed the human health and economic impacts of projected changes in
ozone pollution between 2000 and 2050. They estimated that health costs due to global ozone
pollution above pre-industrial levels by 2050 will be $580 billion and mortalities from acute
exposure will exceed 2 million. Matus et al.(Matus, et al. 2012) found that by improving ozone
and PM pollution, the GDP in China would have increased by U.S. $112 billion (about 5% of
GDP) in 2005. The report released by  OECD estimated the health and economic impacts of global
outdoor air pollution up to 2060 and found that the impacts are especially substantial in Asian
countries(OECD 2016). World Bank also investigated the cost of outdoor air pollution worldwide
and called for action to mitigate air pollution(Worldbank 2016).

With fast economic development and increasing use of fossil fuels, China is faced with

serious air pollution accompanied by severe health problems. Most current studies about health
impacts in China focused on $PM_{10}$ and $PM_{2.5}$ pollution, or ozone pollution in a single city, single
province or at the national level(Zhang, et al. 2006). Few studies try to quantify economic impacts
of ozone pollution at intra-national level. In this study, we focus on health and economic impacts
from ozone pollution at the provincial level in China. Using the daily maximum 8-hour ozone
concentration data provided by the GEOS-Chem model and the latest exposure-response functions
(ERFs), the health-related damages are then integrated into a computable general equilibrium



(CGE) model. In this way, a picture could be drawn on how changes in ozone pollution will affect
health expenditure, labor supply, and the macroeconomy in the Chinese provinces.

# 2  Methods and scenario

## 2.1  General framework

This study develops an integrated approach to consider health and economic impacts of ozone
pollution in China (Figure 1). The integrated framework combines the Asia-Pacific Integrated
Assessment/Computable General Equilibrium (AIM/CGE)-China model, the Greenhouse Gas -
Air Pollution Interactions and Synergies (GAINS)-China model that projects future air pollutant
emissions, an air quality model (GEOS-Chem: version v 10-01; present day: 2008), and a health
impact module.
The AIM/CGE-China model applied in this study can be classified as a multi-sector, multi-
region, recursive dynamic CGE model that covers 22 economic commodities and corresponding
sectors. It includes 30 provinces in China and is solved by the Mathematical Programming System
for General Equilibrium under General Algebraic Modeling System (GAMS/MPSGE) at a one-
year time step(Dai, et al. 2016). The role of the CGE model is (1) to provide energy consumption
data by province and sector to the GAINS model; and (2) to quantify the economic impacts of
health damage. The GAINS-China model provides annual regional emissions data of primary air
pollutants for 30 provinces in China. Note that both the CGE and GAINS models have been
configured extensively to reflect the historical and future pathway of China in reference (Dong, et
al. 2015). For instance, we adjusted the model assumptions to match the historical statistics of



population growth, GDP growth rate, energy (as shown in Figure A3), and air pollutant emissions
(as shown in Figure A4-A8) in each province as much as possible. As for the future, China's GDP
growth and demographic evolution follow the SSP2 (Shared Socio-economic Pathways) scenario
(O'Neill, et al. 2013), which is characterized by moderate economic growth, a fairly rapidly
growing population and lessened inequalities between west, central and east China. An
improvement from the previous study is that, instead of using the concentration results in the
GAINS model, we used the GEOS-Chem model, which is an atmospheric transport and chemistry
model and much better than the simple source-receptor matrix in the GAINS model, to calculate
the daily-maximum-8-hour-average ozone concentration. GEOS-Chem model has been
extensively evaluated and documented in over 100 refereed journal publications(Selin, et al. 2009).
As used here, the model has a horizontal resolution of 0.5 degree latitude and 0.67 degree longitude,
and the meteorological data in 2008 are used for 2030 simulations.

The health module is extended from our previous work(Xie, et al. 2016a; Xie, et al. 2016c)

to quantify the health impacts of ozone pollution and their monetary value. Exposure to
incremental ozone results in health problems called health endpoints, including morbidity and
mortality (all the mortality in this study means ozone-related long-term exposure mortality) (Table
A1 in Supplementary material). The relative risk for the endpoint is believed to be in a linear
relationship with the concentration level(Cakmak, et al. 2016; Kampa and Castanas 2008; Silva,
et al. 2013). When the daily maximum 8-hour ozone concentration is below the threshold value of
70ug/m$^3$, ozone causes no health impacts(Berman, et al. 2012). The method to calculate work loss
time and health expenditure has been described in our previous study(Xie, et al. 2016a). For ozone,
different exposure-response functions from $PM_{2.5}$ are used as shown in Table A1. Annual total
medical expenditure and per capita work loss could be converted from the health impacts and used




as a variation of the household expenditure and labor participation rate in the CGE model that
quantifies the macroeconomic impacts. Furthermore, we also monetize the non-market value of
statistical life lost based on the method(West, et al. 2013; Xie 2011), in which VSLs in all
provinces are calculated using their current GDP per capita values relative to the national average
per capita GDP in 2010 and an income elasticity of 0.5(Viscusi and Aldy 2003). The value of life
ranges from 8.2 to 31.1 million USD(Matus, et al. 2012) in the literature but here we adopt the
latest value of statistical life of about $250,000 USD from empirical investigations using
willingness to pay method in China(Xie 2011).
A more detailed introduction to the health module, AIM/CGE China model, GAINS-China
model and GEOS-Chem model is provided in the Supplemental Material.

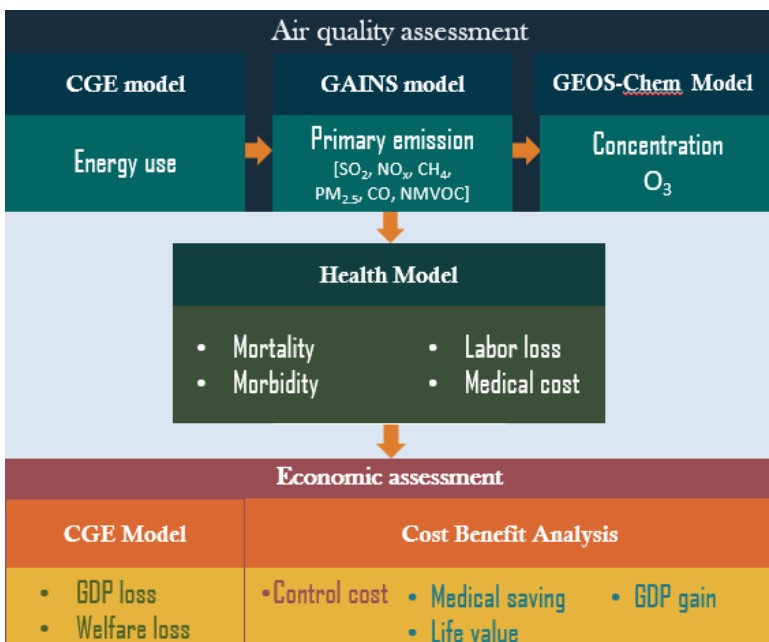


**Figure 1: Integrated research framework of assessing health and economic impacts of**
**air pollution.**



## 2.2 Scenario


Four scenarios are established in this study, namely the reference, woPol, wPol and wPol2
scenarios. More details of the technology settings can be found in the appendix(GAINS-model).
The reference scenario provides the economic results in the CGE model without coupling it
with the health module, which means that ozone pollution related health impacts are ignored. In
other words, ozone pollution causes no additional health service cost, premature death, or work
loss days. The scenario is an ideal situation that does not exist, however, the role of this scenario
is to compare with the other scenarios and evaluate the negative impacts of pollution and benefits
of pollution control.
On the other hand, the remaining three scenarios couple the health module with the CGE
model by considering the health impacts in the CGE model. The woPol scenario assumes that the
penetration rate of mitigation technology is fixed to the 2005 level, implying that the emissions
from additional energy combustion will be uncontrolled in future. It is meant to show the impact
of pollution control policies rather than represents the reality.
By contrast, the wPol scenario takes China's current air pollution policies into account.
Furthermore, the sectoral and provincial differences in emission limit values and time of their
introduction are considered as well. Therefore, various air-pollution-control technologies are used
to reduce pollutant emissions and daily maximum 8-hour ozone concentration to levels below the
woPol scenario. In addition, we also set up a scenario named wPol2, in which more aggressive air
pollutant control technologies are adopted to further reduce the emissions in 2030 of NOx, VOC,
CO by 50% and $CH_4$ by 20% from the wPol scenario.



## 3 Results

### 3.1 Daily maximum 8-hour ozone concentration

The primary emissions of air pollutants (Figure A4 in Supplemental Material) are the same as used in(Xie, et al. 2016a). It can be seen that emissions in the wPol scenario are much lower than in the woPol scenario over all the periods, and emissions in the wPol2 scenario in 2030 are further reduced intentionally. Using these emission pathways as inputs for the GEOS-Chem model the daily maximum 8-hour mean ozone concentration is calculated in 30 provinces of China in both woPol and wPol scenarios in 2030 (Figure 2 (upper two panels)). It shows that ozone concentration is higher in the southwest and northwest i.e., Sichuan (129.1 ug/m3), Qinghai (128.1 ug/m3), and Gansu (115.6 ug/m3) in the woPol scenario, and lower in the East China in both scenarios.

Figure 2 (lower two panels) also shows the changes in daily maximum 8-hour ozone concentration under intensive air pollution control policy (we also show the daily maximum 24-hour concentration in Figure A10). It can be seen that the relationship between reduction in ozone precursors emissions and concentration is not linear. In the wPol scenario, although air pollutants emission reduction is over 50%, daily maximum 8-hour ozone concentration doesn't decrease significantly. The ozone concentration reduction is the most significant in provinces such as Hunan, Anhui, but they only fall by less than 10%. Moreover, there is no significant reduction in Hebei, Shanxi or Inner Mongolia. Conversely, daily maximum 8-hour ozone concentration actually increases in Beijing, Shanghai, and Guangdong in the wPol scenario. Note that we are using the same meteorological data in 2008 and 2030 simulations. Therefore, all the changes are caused by





change in anthropogenic emissions. These patterns, especially the different signs of ozone
concentration changes responding to anthropogenic emissions changes, are resulted from the
different ozone formation regimes these provinces are located at. The great metropolitan regions
such as Beijing, Shanghai, and Guangdong are generally VOC-controlled, and the decreased NOx
emissions in wPol and wPol2 scenarios reduce the ozone destruction rate by reacting with NOx
and thus increase ozone concentrations(Chou, et al. 2011; Xue, et al. 2014).

We also evaluated the impact of anthropogenic emission changes on the 24-hour average

ozone concentrations (Figure A10). The response of 24-hour average ozone concentrations are
significantly different from the daily maximum 8-hour, with the former has percentage changes
toward the positive axis. At regions with increased concentrations, the changes in concentrations
are more prominent if we use a 24-hour average matric than the daily maximum 8-hour, however,
at regions with decreased concentrations (such as Beijing, Shanghai, and Guangdong), the
magnitude of changes becomes less significant. This pattern is largely associated with the diurnal
cycles of ozone formation and removal. While the daily maximum 8-hour mainly represents the
daytime when active ozone production is occurring, the 24-hour average is also influenced by the
nighttime condition when photochemical ozone formation ceased, and anthropogenic NOx
emissions efficiently destruct ozone. Therefore, decreasing anthropogenic emissions largely
increase nighttime ozone concentrations(Zhang, et al. 2004).

To elucidate the portion of natural contribution to ozone formation, we conducted an

additional simplified experiment in the GEOS-Chem model by reducing the human-related
emissions to zero and calculating the resulting daily maximum 8-hour ozone concentration. This
concentration is defined as natural background in our case. Figure 3 shows that the provinces could
be divided into three groups based on the percentage of ozone from natural sources in the wPol



scenario. The first group is natural source-dominated provinces where human activity source is
lower than 20%, including Xinjiang, Hainan, Qinghai, Gansu, Tianjin, Shanghai and Inner
Mongolia. In these provinces that are home to tens of millions of people, daily maximum 8-hour
ozone concentration reduction in wPol scenario is not significant, implying that the health damage
caused by ozone pollution is hard to mitigate by policy intervention. The second group is where
the human activity source is between 20% to 40%, including Beijing, Hebei, Shanxi, Liaoning,
Jilin, Jiangsu, Heilongjiang, Shandong, Henan, Guangdong, Guangxi, Sichuan, Yunnan, Shaanxi
and Ningxia. In the third group, anthropogenic emissions dominate (>40%), including Zhejiang,
Anhui, Fujian, Jiangxi, Hubei, Hunan, Chongqing and Guizhou. Daily maximum 8-hour ozone
concentration could decrease a lot in these provinces by cutting the anthropogenic emissions in the
wPol scenario(Zhang, et al. 2004).



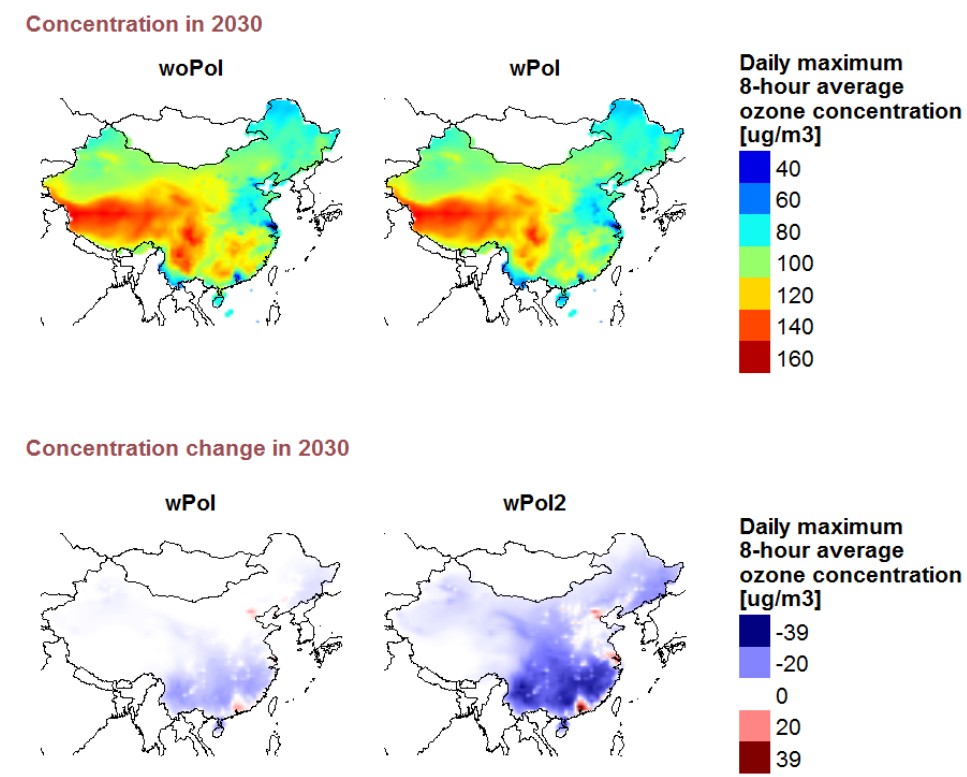


**Figure 2: Daily maximum 8-hour ozone concentration in woPol and wPol scenarios**

**(upper) and change from woPol to wPol and wPol2 scenarios (lower).**





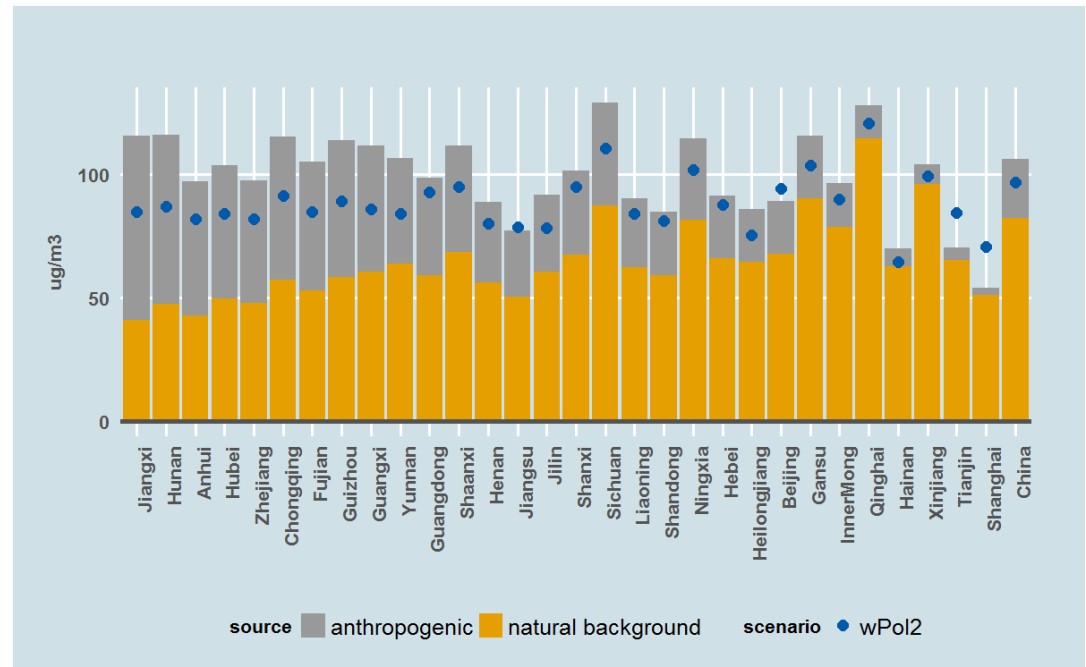


**Figure 3: Daily maximum 8-hour ozone concentration from natural background, anthropogenic emissions in the woPol and wPol2 scenarios in 2030.**



## 3.2 Health impacts


Health endpoints from ozone pollution include mortality, morbidity and work loss days. In
this study, the morbidity consists of coughs, asthma, bronchodilator usage, lower respiratory
symptoms, and respiratory-related hospital admissions (Table A1). As Figure 4 shows, we quantify
the opportunities to obtain ozone-related diseases, premature death and payment for these kinds of
illness for people exposed to two levels of daily maximum 8-hour ozone concentration in 30
Chinese provinces.
In the woPol scenario, the daily maximum 8-hour ozone concentration in most parts of China
will be still above the standard level of 70 ug/m$^3$ in 2030(Amann 2008; Bickel and Friedrich 2004).





Only Hainan (66 ug/m$^3$) and Shanghai (66 ug/m$^3$) would be able to meet the national standard,
while in the populous regions like Beijing (96.1 ug/m$^3$), Tianjin (77.8 ug/m$^3$), and Jiangsu (75.1
ug/m$^3$), daily maximum 8-hour ozone concentration is high enough to cause various health impacts
as shown in Figure 4 (left column), including mortality, per capita morbidity, per capita work loss,
per capita health expenditure and value of life lost (VOLL). We also calculate the mitigation
benefit (Figure 4 right) from air pollution control policy in the wPol scenario (Figure 4 right
column).
In 2030, the national total number of mortality is about 583.0 (95% confidence interval 230.8-
1189.5) thousand person in woPol scenario. In wPol and wPol2 scenario, mortality is 491.0 (209.2-
1078.4) and 335.3 (169.3-872.5) thousand persons. Air pollution control in the wPol scenario
could lead to a decrease in mortality by 92.04 thousand persons. At the provincial level, Sichuan,
Gansu, Shaanxi and Hunan encounter most of the ozone-related mortality in the woPol scenario,
about 73.6 (24-123) 17.7 (5.8-30), 23.5 (7.6-39) and 47 (15-78) thousand person per year,
respectively. In the wPol scenario, the majority reduction in mortality takes place in provinces in
South China such as Hunan (12,000 people), Guangxi (9,000 people), Jiangxi (8,100 people) and
Yunnan (7,100 people). However, even in this scenario, the total number of mortality in these four
provinces is 140,100 people (46-235), accounting for 61.2% of national ozone-related mortality.
As indicated by per capita morbidity, the provinces in the west and central China such as
Sichuan, Qinghai, Jiangxi, Hunan and Chongqing, with higher daily maximum 8-hour ozone
concentration, will be severely impacted. People in these provinces run a higher risk, about 4-5%,
of suffering from health effects such as asthma attacks, respiratory hospital admission, coughs,
and mortality from ozone exposure. In contrast, provinces in the east of China with lower daily



maximum 8-hour ozone concentration, such as Tianjin, Jiangsu, Beijing and Shandong, are at a
lower risk, about 1-2 %, of suffering from such health effects from ozone exposure.

Ozone exposure also leads to work loss days. Premature deaths among those aged between

15 to 65 years old will reduce labor supply and total work time. However, there is no concentration-
response function about work loss days for ozone exposure in the literature. In this study, we
converted minor restricted activity days of ozone into work loss days based on the relationship of
PM2.5, e.g., minor restricted activity days are 2.78 times work loss days. Figure 4 shows the per
capita work loss hours due to morbidity and cumulative mortality. In 2030, the national average
per capita work loss is 1.24 (0.6-2.48), 1.06 (0.55-2.28) and 0.79 (0.46-1.92) hours respectively in
the woPol, wPol and wPol2 scenarios in China. At the provincial level, Qinghai, Sichuan, Gansu
and Xinjiang encounter more work loss hours, about 2.5 (0.97-4.05), 2.5 (0.97-4.03), 1.93 (0.75-
3.12) and 1.46 (0.57-2.23) hours respectively in the woPol scenario. The recovered work loss in
the wPol scenario ranges from 0.39 hour in Yunnan Province to -0.24 (increase) hour in Beijing.





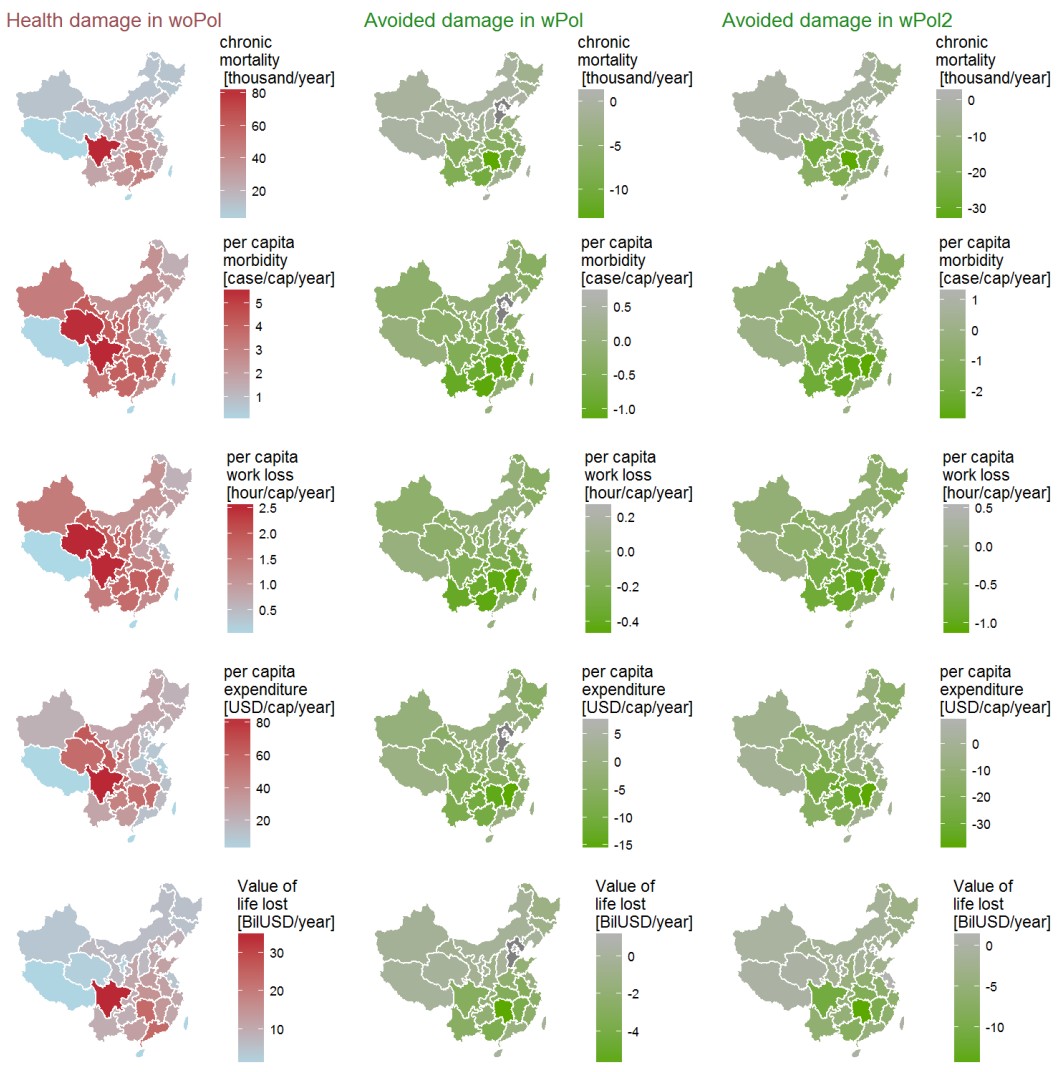


**Figure 4: Health damage due to ozone pollution (left/red) and benefit of mitigation in 2030 (right/green).**

## 3.3 Economic impacts

Figure 4 (the bottom two rows) and Figure 5 show the economic loss due to ozone-related health impacts, including health expenditure, value of life lost, GDP loss and welfare loss.




**Medical expenditure**. The total health expenditure on ozone exposure-related morbidity in
China in 2030 is estimated to be 38.25, 32.25 and 21.50 billion USD (2002 constant price) in the
woPol, wPol and wPol2 scenarios, equivalent to per capita expenditure of 28.03, 23.64 and 15.75
USD, respectively. The top five provinces account for most of the health expenditure in the woPol
scenario, such as Sichuan (7.33 billion USD), Gansu (1.77 billion USD), Hunan (3.97 billion
USD ), Jiangxi (2.56 billion USD ) and Shaanxi (1.33 billion USD). The top 3 provinces with
highest per capita expenditure in the woPol scenario are different; the highest province is still
Sichuan (84.1 USD), followed by the relatively poor provinces of Qinghai (56.4 USD), and Gansu
(65.3 USD). This implies that ozone pollution could become a non-ignorable economic burden to
the low-income residents in the west of China. Simultaneously, in the wPol scenario reduction
rates of total expenditure are as follows in these five provinces: -8.67% (Sichuan), -6.9% (Gansu),
-8.26% (Hunan), -25.54% (Jiangxi), -27.62% (Shaanxi).
**GDP loss and welfare loss**. Both labor supply loss and medical expenditure increase will
affect macroeconomic indicators such as GDP and residential welfare. As indicated in Figure 5, in
2030, China will experience a GDP loss of about 0.034% (0.028~0.062%) in the woPol scenario,
0.030% (0.025~0.055%) in the wPol scenario and 0.024% (0.023~0.045%) in the wPol2 scenario.
At the provincial level, provinces in the west and southwest will experience higher GDP losses,
for example, Qinghai (0.084%, 0.080% and 0.074% respectively in the woPol, wPol and wPol2
scenarios), Sichuan (0.082%, 0.076% and 0.060%), Gansu (0.067%, 0.062% and 0.51%), Ningxia
(0.065%, 0.062% and 0.047%), and Hunan (0.066%, 0.051% and 0.028%). By contrast, Shanghai
and Hainan will experience much lower GDP loss from ozone pollution, about 0.001% in the
woPol (0.006% in the wPol scenario). GDP loss is 0.030% (0.038%) in Beijing, 0.006% (0.0038%)
in Tianjin, and 0.013% (0.011%) in Jiangsu.



Provinces with larger GDP, population and high ozone concentration display the following
absolute values of GDP loss. The higher absolute value of GDP loss is 430.5 (378.0~794.3) billion
CNY in Guangdong, 315.1 (259.2~572.3) billion CNY in Sichuan, 257.7 (194.2~459.9) billion
CNY in Zhejiang, 188.0 (143.1~335.8) billion CNY in Hunan, and 181.2 (167.8~338.9) billion
CNY in Shandong.
Welfare loss is defined as total consumption change, which is measured by Hicks' equivalent
variation(Fujimori, et al. 2015). In China, welfare loss from ozone-related health impacts in 2030
is about 0.048% and 0.042% and 0.033% respectively in the woPol, wPol and wPol2 scenarios.
Welfare loss is higher in provinces such as Qinghai (0.15%, 0.14% and 0.13%), Ningxia (0.14%,
0.13% and 0.11%), Sichuan (0.12%, 0.11% and 0.76%) in the woPol, wPol and wPol2 scenarios
in 2030. These provinces are in the west of China, where ozone from natural sources is quite high.
The difference between the two scenarios is not significant.
**Value of statistical life lost**. Figure 5 also shows the willingness to pay for each health
endpoint. The benefits of avoided air pollution mortality and morbidity are monetized using value
of statistical life (VSLs). In 2030, the national value of statistical life lost is about 284.77 and 241.5
billion USD respectively in the woPol and wPol scenarios, which is about 2.34% and 1.99% of
GDP. At the provincial level, VSLs of Sichuan, which has the highest mortality and moderate per
capita GDP, is the highest (34.44 billion USD, or 7.37 % of GDP in woPol), followed by the
western provinces of Gansu (6.92 billion USD, or 5.72 % of GDP), Xinjiang 3.97 billion USD, or
2.95 % of GDP), and Shaanxi 10.84 billion USD, or 4.88 % of GDP).






**Figure 5: GDP loss, welfare loss and value of statistical life lost due to ozone pollution in 30 provinces in 2030.**





## 306  4 Discussion

Daily maximum 8-hour ozone concentrations within China vary by region and by season.
The contribution of anthropogenic and biogenic emissions to regional-scale ozone concentration
also varies by region. Ozone concentrations are higher in the west of China but lower in the east,
and higher in summer (due to higher active reaction of photochemical production) but lower in
winter in most provinces and cities, dominated by natural background ozone in some provinces
while by anthropogenic sources elsewhere (Figure A9). When national and local governments
launch new air pollution control policies, they should consider these features.
In accordance with the features of ozone concentration distribution, ozone-related health
impacts are more severe in the western provinces with higher daily maximum 8-hour ozone
concentration and moderate population density. The provinces of Qinghai, Sichuan, Gansu and
Jiangxi suffer from higher per capita morbidity, more work hour loss and higher economic impacts.
In contrast, health impacts are lower in the east of China, where the population density is much
higher than west. The provinces in the southwest and northwest experience higher GDP loss and
welfare loss due to ozone pollution. At the same time, these provinces are relatively less developed
and have less motivation to control ozone pollution. Considering this regional variation, the
government should make specific ozone control strategies for different regions. Moreover, since
ozone is long-range transboundary air pollution, to control ozone pollution effectively,
collaboration among provinces is an imperative.
We find that it is more difficult to reduce daily maximum 8-hour ozone concentration
compared with $PM_{2.5}$ (Figure A4) because the ozone generation process is not in a linear
relationship with precursor emissions, implying that in the longer term, ozone pollution will be a



more persistent air pollution problem in China. Although ozone precursor emissions have been
reduced a lot from the woPol to wPol and wPol2 scenarios (Figure A4), the daily maximum 8-
hour ozone concentration reduction is very limited (less than 10%) in the wPol scenario. Even
more aggressive reduction efforts are made in the wPol2 scenario, in contrast to $PM_{2.5}$ whose daily
concentration could be reduced by over 70% in almost all provinces, reduction rates of daily
maximum 8-hour ozone concentration are merely around 20% in most provinces. Conversely, in
urban areas around Beijing, Shanghai and Guangzhou, it actually increases. A similar phenomenon
has been reported in previous studies in China. For instance, Chou et al.(Chou, et al. 2011) found
that the mixing ratio of ozone increased with the increasing $NO_2/NO$ ratio, whereas the $NO_z$ mixing
ratio leveled off when $NO_2/NO > 8$ (Chou, et al. 2011). Consequently, the ratio of ozone to $NO_z$
increased to above 10, indicating the shift from a VOC-sensitive regime to a $NO_x$-sensitive regime.
Xue et al. found varying and considerable impacts of ozone generation processes in different areas
of China depending on the atmospheric abundances of aerosol and $NO_x$(Xue, et al. 2014). This is
partly owing to the fact that most of $PM_{2.5}$ is from anthropogenic activities like industry and
transportation, while relatively less is from natural sources, such as desert, farmland, forest burning
and sea salt. But for ozone, a significant source is natural emissions, which are beyond the control
of human activity. One study from WHO shows human exposure to ozone during the winter is
reduced because more time is spent indoors. Building structures and slow rates of ventilation
reduce ozone penetration indoors even during the summer(Amann 2008). Therefore, the
government should provide daily public information about air quality, and the public should adjust
their lifestyles according to the air quality information.

As we know, China is suffering from extremely serious $PM_{2.5}$ pollution. Air pollution control

policy can reduce $PM_{2.5}$ pollution as well as ozone pollution. Air pollution control policy that can



reduce primary emissions reduction such as NOx, SOx, VOC leads to reduce both PM$_{2.5}$ pollution
and ozone pollution. In such a case, controlling PM$_{2.5}$ pollution also brings the benefit of reduction
of ozone pollution. We also compared the national impacts of ozone and PM$_{2.5}$ pollution in China
(Figure 6, PM$_{2.5}$ results are updated from(Xie, et al. 2016b)). Regarding exposure-response
functions, ozone has smaller ERFs than PM$_{2.5}$, including the mortality and most morbidity. For
PM$_{2.5}$, the World Health Organization (WHO) standard is 10 ug/m$^3$, emissions above which will
lead to health effects. For ozone, the threshold value is 35ppbv (equivalent to 70 ug/m$^3$). PM$_{2.5}$
concentration is much higher in high population density areas, while daily maximum 8-hour ozone
concentration is higher in relatively low populated western provinces. As a result, it is found that
health and economic impacts from ozone are much smaller than PM$_{2.5}$ pollution except for per
capita morbidity and expenditure. Taking the wPol scenario in 2030 for example, total mortality
is 0.49 million due to ozone whereas it is 2.43 million due to PM$_{2.5}$. Per capita work loss is only
1.1 hour from ozone while it is 11.4 hours due to PM$_{2.5}$. Conversely, upper respiratory symptoms
dominate PM-related endpoints while the overwhelming endpoints related to ozone are
bronchodilator usage and weaker respiratory symptoms. Per capita morbidity caused by ozone (2.3
times per capita per year) is more than 20 times higher than PM$_{2.5}$ (0.1 times per capita per year)
mainly due to bronchodilator usage, consequently, per capita expenditure due to ozone is 23.64
USD, which is much higher than that caused by PM$_{2.5}$ (4.82 USD). Furthermore, ozone causes less
GDP loss (0.03%) than PM$_{2.5}$ (0.36%-0.83% in the wPol scenario and 1.14-2.82% in the woPol
scenario as reported in(Xie, et al. 2016a)). Moreover, the GDP loss due to both PM$_{2.5}$ and ozone
pollution in the woPol scenario in our study is comparable to that reported by the OECD[14] (2.6%
in 2060). Matus et al. used a CGE model to estimate the benefits of air pollution control in the
USA, and found that the benefits rose steadily from 1975 to 2000 from 50 billion USD to 400



billion USD (from 2.1% to 7.6% of market consumption)(Matus, et al. 2008). This result is also
comparable with our result. US EPA's study also shows that the benefit from The Clean Air Act
is higher than the cost in the US(Agency 1999). On the other hand, if the air quality is improved,
fewer people die due to air pollution. The population projection and energy consumption will also
be changed in the future. If considering this feedback in CGE model, the GDP loss and welfare
loss would be different.

Uncertainty within our framework could be classified into three sources. The first source is

uncertainty of future economic development and energy consumption in the CGE model. The
second source is estimation of future air pollutant emissions and ozone concentration, which is
related to both technology selection and the behavior of the GEOS-Chem model. The last source
is related to ERFs used in the health model. In terms of uncertainty of ERFs, the numbers in the
parenthesis show 95% CI of ERFs.

Despite our pioneering efforts in quantifying the health and economic impacts of ambient

ozone and $PM_{2.5}$ pollution, there are some limitations and uncertainties, which need further
investigation. Many epidemiological studies show exposure to higher ozone concentration not only
leads to health problem, but also leads to reductions in the amount of effective labor, which is
measured in labor productivity(Brauer, et al. 1996; Korrick, et al. 1998). But the effects on
productivity cannot be quantified in this study. If we consider these kinds of impacts, the economic
impact from ozone pollution will be higher than current results. Besides, our results may be
underestimated because we neglect mortality among those younger than 30, including effects on
children and neonatal effects(West, et al. 2013). Furthermore, as noted in the supplementary
information, there are no ERFs for work loss days for ozone, and as the second best approach we
converted it from the restricted activity day, which leads to uncertainties concerning the



quantifying of the market economic impacts in the CGE model. We expect future epidemic studies
could fill this gap.






**Figure 6: Comparing health impacts between PM$_{2.5}$ and ozone.**

# 5 Acknowledgement

This study was supported by the Environmental Research and Technology Development Fund
(S-12-2 and 2-1402) of the Ministry of the Environment, Government of Japan and the Natural
Science Foundation of China (71704005). The authors are grateful for the comments from the
anonymous reviewers of this paper.

# 6 Appendix

## 6.1 The health assessment module

**Health endpoint**

The health module is extended from our previous work (Xie, et al. 2016a)to quantify the
health impacts of ozone pollution and monetize the value of such health impacts. Exposure to
incremental ozone leads to health problem called health endpoints, which are categorized into
morbidity and chronic mortality (Table A1). This study follows the methodology that the relative
risk (RR) for the endpoint is in a linear relationship with the concentration level(Cakmak, et al.
2016; Kampa and Castanas 2008; Silva, et al. 2013). As showed in Health equation 1, when the
concentration is lower than the threshold value of 70ug/m$^3$ (Amann 2008; Bickel and Friedrich
2004) RR is 1, there is no concentration-response function(CRF) to quantify health impacts of
ozone. Linear function assumes that the CRF is a constant (Health equation 2). The number of



health endpoints is estimated by multiplying RR with population and reported cause-specific
mortality rate.

All results are region $r$, year $y$, scenario $s$, and uncertainty range $g$ specific. For simplification,

they are omitted in the following description.
**Health equation (1):**
$$RR_{p,r,s,y,m,e,v}(C) = \begin{cases} 1, & \text{if } C_{p,r,s,y} \leq C0_p \\ 1 + CRF_{m,e,v} \times (C_{p,r,s,y} - C0_p), & \text{linear function,} \quad \text{if } C_{p,r,s,y} > C0_p \end{cases}$$
**Health equation (2):**
$$EP_{p,r,s,y,m,e,v}(C)$$
$$= \begin{cases} P_{r,y,m} \times (RR_{p,r,s,y,m,e,v}(C) - 1), & \text{for linear morbidity function} \\ P_{r,y,m} \times (RR_{p,r,s,y,m,e,v}(C) - 1) \times I_{r,\text{"all cause"}}, & \text{for linear mortality function} \end{cases}$$

where

RR(C): Relative risk for endpoint at concentration C [case/person/year or day/person/year]
EP: Health endpoint [case/year or day/year]
C: Concentration level of pollutant
C0: Threshold concentration that causes health impacts (10 ug/m3 for $PM_{2.5}$ and 70 ug/m3 for
ozone.)
CRF: Concentration-response function
P: Population, aged 15-65 for work loss day, age 25-65 for Ischemic heart disease and Stroke,
and entire cohort for other endpoints
$\hat{I}$: (cause-specific mortality rate
I: Reported average annual disease incidence (mortality) rate for endpoint
$I_{r,\text{"all cause"}}$: Reported average annual natural death rate for endpoint
$\alpha, \gamma, \delta$: Parameters that determine the shape of the non-linear concentration-response
relationship for chronic mortality.
Suffix p, r, s, y, m, e, v represent pollutant ($PM_{2.5}$ and $O_3$), region, scenario, year, endpoint
category (morbidity or mortality), endpoint, value range (medium, low and high), respectively.

**Annual per capita work loss rate**



Annual total work loss day (WLD) of a region is a summation of work loss day from
morbidity and cumulative work loss day from chronic mortality aged from 15 to 65 years old
(Health equation 3). Based on death rates for different age group and cause-specific mortality from
China health statistics, we assume 4% of total chronic mortality is aged between 15 and 65 years
old. Note that different from $PM_{2.5}$ impact, in the literature there is no ERF for work loss day for
ozone pollution. In this study, we converted minor restricted activity day of ozone into work loss
day based on the relationship of $PM_{2.5}$, e.g., minor restricted activity day is 2.78 times of work loss
day. Annual per capita work loss rate (WLR) is obtained by dividing WLD with working
population and annual working days (Health equation 4). In the CGE model, WLR is used to
calculate the actual labor force after subtracting the work loss (Health equation 5).
**Health equation (3):**
$$WLD_{p,r,s,y,v} = \sum_{m} (EP_{p,r,s,y,m,"wld",v}) + \sum_{e,y'<y} (EP_{p,r,s,y',"mt",e,v}) \times 0.04 \times DPY$$

**Health equation (4):**
$$WLR_{p,r,s,y,v} = \frac{WLD_{p,r,s,y,v}}{DPY \times P_{r,y,"15-65"}}$$

**Health equation (5):**
$$LAB_{p,r,s,y,v} = LAB0_{r,"ref",y} \times (1 - WLR_{p,r,s,y,v})$$

Where
**WLD**: Annual work loss day [day/year]
**WLR**: Annual per capita work loss rate
**"wld"**: Subset "Work loss day" of e
**"mt"**: Subset "Chronic mortality" of m
**LAB**: Labor force after considering work loss


**LAB0**: Labor force in the reference scenario
**DPY**: Per capita annual working days (5 day/week * 52 week/year = 260 day/year)

**Health expenditure**

Additional health expenditure is obtained by multiplying outpatient and hospital admission

price with total endpoints (Health equation 6). The price is a function of per capita GDP of each
province (Health equation 7), and the parameters $\beta, \theta$ are estimated through regression analysis
of statistical price by disease and GDP of each province from 2003 to 2012. Additional medical
expenditure is regarded as household expenditure pattern change, which means as more money is
spent on medical services, less is available on other commodities.

**Health equation (6):**

$$HE_{p,r,s,y,m,e,v} = PR_{r,s,y,e,v} \times EP_{p,r,s,y,m,e,v}$$

**Health equation (7):**

$$PR_{r,s,y,e,v} = \beta_{r,e} \times GDPPC_{r,s,y} + \theta_{r,e}$$
Where:

**HE**: Total additional health expenditure [billion Yuan/year]
**PR**: Price of medical service [Yuan/case]
$GDPPC_{r,s,y}$: Per capita Gross Domestic Production from CGE model
$\beta_{r,e}, \theta_{r,e}$: Parameters derived from regression analysis of medical service price



Table A1: Concentration response functions from ozone related health endpoints.

| Category | Endpoint | Unit | Medium | C.I. (95%) Low | C.I. (95%) High |
|---|---|---|---|---|---|
| Morbidity | Work loss day | day/person/ug-m3 | 0.004126 | 0.001655 | 0.006627 |
| | Minor restricted activity day | day/person/ug-m3 | 0.0115 | 0.0044 | 0.0186 |
| | Asthma attacks | case/person/ug-m3 | 0.00429 | 0.00033 | 0.00825 |
| | Respiratory hospital admissions | case/person/ug-m3 | 0.000004 | 0.000001 | 0.000006 |
| | Bronchodilator usage | case/person/ug-m3 | 0.073 | -0.0255 | 0.157 |
| | Lower respiratory symptoms | case/person/ug-m3 | 0.016 | -0.043 | 0.081 |
| Mortality | Chronic mortality | case/person/ug-m3 | 0.002 | 0.00065 | 0.00335 |
| | Value of statistical life | million USD/life | 0.25 | | |

Source(Amann 2008; Apte, et al. 2015; Bickel and Friedrich 2004; Cao, et al. 2011)
**Value of statistical life**
As showed in Health equation 8, we also quantified the value of statistical life (VSL) using
the willingness to pay approach following the method(West, et al. 2013). The national average
willingness to pay for avoiding premature death is 250 thousand USD (Table A1)(Xie 2011). VSLs
in all provinces are calculated using their current GDP per capita values relative the national
average per capita GDP in 2010 and an income elasticity of 0.5(Viscusi and Aldy 2003).
**Health equation (8):**
$$VoE_{p,r,s,y,e} = WTP_{r,y,e} \times EP_{p,r,s,y,m,e,v} \times (\frac{GDP_{r,y}}{GDP_{\text{"China","2010"}}})^{0.5}$$
Where:
•   $VoE_{p,r,s,y,e}$: Value of health endpoint;
•   $WTP_{r,y,e}$: Willingness to pay for avoiding premature death and morbidity.



## 6.2 The CGE model


The global CGE model applied in this study can be classified as a multi-sector, multi-region,
recursive dynamic CGE model that covers 22 economic commodities and corresponding sectors,
and eight power generation technologies as detailed in Table A2. As a special model feature, the
number of modelling regions, both international and within China is highly flexible to allow for a
wide range of studies. In this regard 3, 7, or 30 provincial units of China and 1, 3, or 14 international
regions could be analyzed consistently, as summarized in Table A3. This CGE model is solved by
Mathematical Programming System for General Equilibrium under General Algebraic Modeling
System (GAMS/MPSGE)(Rutherford 1999) at a one-year time step. The model includes a
production block, a market block with domestic and international transactions, as well as
government and household incomes and expenditures blocks. Activity output for each sector
follows a nested constant elasticity of substitution (CES) production function. Inputs are
categorized into material commodities, energy commodities, labor, capital and resources.
Technical descriptions have been introduced in(Dai, et al. 2015; Xie, et al. 2016a). It has been used
widely for assessing China's climate mitigation at the national(Dai, et al. 2011; Dai, et al. 2012)
and provincial level(Cheng, et al. 2016; Cheng, et al. 2015; Tian, et al. 2016; Wu, et al. 2016; Xie,
et al. 2016a). The model has been configured extensively to reflect the historical and future
pathway of China in reference (Dong, et al. 2015). For instance, we adjusted the model
assumptions to match the historical statistics of population growth, GDP growth rate, energy (as
shown in Figure A1) and air pollutant emissions (as shown in Figure A2) in each province as much
as possible. As for the future, China's GDP growth and demographic evolution follows SSP2
(Shared Socio-economic Pathways) scenario (O'Neill, et al. 2013), which is characterized by



moderate economic growth, fairly rapid growing population and lessened inequalities between
west, central and east China.




*Table A2 Classification of sectors in the model*

| Nr. | Code | Note | Nr. | Code | Note |
|---|---|---|---|---|---|
| 1 | Cagri | Agriculture | 16 | COthManuf | Other manufacturing |
| 2 | Coal | Coal | 17 | Celec | Power generation |
| 3 | Coil | Crude oil | 18 | CGas | Manufactured gas |
| 4 | Cmin | Other Mining | 19 | Cwater | Water production |
| 5 | CFdTbc | Food and Tabaco | 20 | CCnst | Construction |
| 6 | CTxt | Textile | 21 | CTrsp | Transport |
| 7 | Cpaper | Paper | 22 | Csvc | Service |
| 8 | Cpet | Petrol oil | i | CoalP | Coal power |
| 9 | Cchem | Chemicals | ii | CoilP | Crude oil power |
| 10 | CNonMPrd | NonMetal product | iii | Cngs | Natural gas power |
| 11 | CMetSmlt | Metal smelting and processing | iv | Hydro | Hydro power |
| 12 | CMetPrd | Metal product | v | Nuclear | Nuclear power |
| 13 | CMchn | Machinery | vi | Wind | Wind power |
| 14 | CTspEq | Transport equipment | vii | Solar | Solar power |
| 15 | CElcEq | Electronic equipment | viii | Biomass | Biomass power |






*Table A3 Model regions defined in the CGE model*

| | **China regions** | | | **International regions (excl. China)** |
|---|---|---|---|---|
| *30 provinces in China* | *3 China-Regions* | *7 China-Regions* | | *14 International Regions* |
| Beijing | East | North China | AFR | Africa |
| Tianjin | East | North China | AUS | Australia-New Zealand |
| Hebei | East | North China | CAN | Canada |
| Shanxi | Central | North China | CSA | Central and South America |
| Inner Mongolia | West | North China | EEU | Eastern Europe |
| Liaoning | East | Northeast China | FSU | Former Soviet Union |
| Jilin | Central | Northeast China | IND | India |
| Heilongjiang | Central | Northeast China | JPN | Japan |
| Shanghai | East | East China | MEA | Middle East |
| Jiangsu | East | East China | MEX | Mexico |
| Zhejiang | East | East China | ODA | Other Developing Asia |
| Anhui | Central | East China | SKO | South Korea |
| Fujian | East | East China | USA | United States |
| Jiangxi | Central | Central China | WEU | Western Europe |
| Shandong | East | East China | | |
| Henan | Central | Central China | | *3 International Regions* |
| Hubei | Central | Central China | NON-OECD | Non-OECD countries |
| Hunan | Central | Central China | OECD | OECD countries |
| Guangdong | East | South China | BRICS | Brazil, Russia, India and South Afric |
| Guangxi | West | South China | | |
| Hainan | East | South China | | *1 International Region* |
| Chongqing | West | Southwest China | ROW | Rest of the world |
| Sichuan | West | Southwest China | | |
| Guizhou | West | Southwest China | | |
| Yunnan | West | Southwest China | | |
| Shaanxi | West | Northwest China | | |
| Gansu | West | Northwest China | | |
| Qinghai | West | Northwest China | | |
| Ningxia | West | Northwest China | | |
| Xinjiang | West | Northwest China | | |
| Tibet | West | Southwest China | | |



**Production**


For each sector (j) in region (r), gross output $Q_{r,j}$ is produced using inputs of labor ($L_{r,j}$),
capital ($K_{r,j}$), energy ($E_{r,j}$ $Coal_{r,j}$, $oil_{r,j}$, $gas_{r,j}$ and $ele_{r,j}$), and non-energy material ($M_{r,j}$). In
some sectors (Cagri, Coal, Coil, Cmin), resource ($RES_{r,j}$) is also input. A five-level nested function
is used to characterize the production technologies as showed in *Figure* and CGE Equation 2
below. The producer maximizes its profit by choosing its output level and inputs use, depending
on their relative prices (CGE Equation 1) subject to its technology (CGE Equation 2).
**CGE Equation (1):**
$$\max \pi_{r,j} = p_{r,j} \cdot Q_{r,j} - \left( \sum_{i=1}^{N} p_{r,i} \cdot X_{r,i,j} + \sum_{v=1}^{V} \omega_{r,v} \cdot V_{r,v,j} \right) - T_{r,j}^{z}$$
Subject to the production technology:
**CGE Equation (2):**
$Q_{r,j} =$
$LEO_{1rj}\{M_{r,i,j}, RES_{r,j}, CES_{2vae}(CES_{3va}(K_{r,j}, L_{r,j}, CES_{3e}(ele_{r,j}, CES_{4fos}\langle coal_{r,j}, gas_{r,j}, oil_{r,j}, pet_{r,j}\rangle)))\}$
Where
$\pi_{r,j}$    is the profit of j-th producers in region r;
$Q_{r,j}$    Output of j-th sector in region r;
$X_{r,i,j}$    Intermediate inputs of i-th goods in j-th sector in region r; As shown in *Figure A1*, $X_{r,i,j}$
includes $M_{r,i,j}$ (non-energy material), $ele_{r,j}$ (electricity), $coal_{r,j}$ (coal), $gas_{r,j}$ ( natural gas or
manufactured gas), $oil_{r,j}$ (crude oil), $pet_{r,j}$ (refined oil) and $RES_{r,j}$ ( resource which is originated from
the natural resource endowment);
$V_{r,v,j}$    v-th primary factor inputs in j-th sector in region r;
$p_{r,j}$    Price of the j-th composite commodity;
$\omega_{r,v}$    v-th factor price in region r;
$K_{r,j}$    is capital input in sector j;
$L_{r,j}$    is labor force in sector j;
$T_{r,j}^{z}$ is production tax in sector j; $T_{r,j}^{z} = p_{r,j} \cdot Q_{r,j} \cdot \tau_{r,j}$, *where $\tau_{r,j}$ is the production tax*
*rate*;



$CES_{krj}$ is the CES function at the k-th nesting level, the first level, $LEO_{1rj}$, is Leontief
function, the second level ($CES_{2vae}$) is aggregation of value added and energy composite, the
third level $CES_{3va}$ is aggregation of value added, and $CES_{3e}$ is aggregation of energy
composite, the fourth level $CES_{4fos}$ is aggregation of fossil energy inputs.
The following conditions apply in this regard:
Land inputs are considered only for agriculture sector (Cagr), other resources are considered
for crude oil and natural gas extraction (Coil), coal mining (Coal) and other mining (Cmin) sectors;
Within energy transformation sectors such as oil refining (Cpet), gas manufacturing (Cgas),
primary energy commodities are considered as material inputs;
The power sector is modelled by three fossil-fired (coal, gas and oil) and five non-fossil
(nuclear, hydro, wind, solar and biomass) technologies( *Figure b)*. The energy bundle is not
combined with capital for fossil-fired technologies, but linked directly to activity output. This
means that electricity output is in a linear relationship with energy inputs.
Labor is assumed to be fully mobile across industries within a region but immobile across
regions. The mobility feature of capital follows a putty-clay approach, which means that vintage
capital is immobile across either regions or industries while new investment is fully mobile across
industries within a region.


*Figure A1 Nesting of production structure*



*a, except for electricity sector; b, electricity sector. σ is elasticity of substitution for inputs.*
$VAE_{r,j}, VA_{r,j}, E_{r,j}, FOS_{r,j}$ *are CES composites of value added & energy, value added,*
*energy and fossil energy, respectively.*

**Final demand**

Household and government sectors are represented as two different final demand sectors. As
CGE Equation 3 shows, the representative household receives income from the rental of primary
factors ($\sum_{v=1}^{V}(\omega_{r,v} \cdot V_{r,v}) + \sum_{j}(pld_r \cdot QLAND_{r,j}) + \sum_{s,j}(p_r^{res} \cdot QRES_{r,j,s})$) and lump-sum
transfer from the government. The income is used for either investment (or saving, $T_r^p$) or final
consumption ($\sum_{i} p_{r,i}^q \cdot X_{r,i}^p$). Households maximize their utility by choosing the levels of final
consumption of commodities, subject to the constraints of their income and commodity prices (see
the income balance in CGE Equation 3). Total investment is assumed exogenously by CGE
Equation 12. On the other hand, the government collects taxes ($T_r^d + \sum_j T_{r,j}^z + \sum_j T_{r,j}^m$) and
spends the tax revenue in providing public services ($p_r^i \cdot x_{r,i}^g$) as explained in CGE Equation 4. The
demands ($DEM_r^d$) of household consumption, investment goods and government are specified
using Cobb-Douglas utility or demand functions (CGE Equation 5).
**CGE Equation (3): income balance of the representative household**
$$\sum_{v=1}^{V}(\omega_{r,v} \cdot V_{r,v}) + \sum_{j}(pld_r \cdot QLAND_{r,j}) + \sum_{s,j}(p_{r,s}^{res} \cdot QRES_{r,j,s}) - T_r^d = \sum_{i} p_{r,i}^q \cdot X_{r,i}^p + S_r^p$$
**CGE Equation (4): income balance of the government**
$$T_r^d + \sum_{j} T_{r,j}^z + \sum_{j} T_{r,j}^m = \sum_{i} p_{r,i} \cdot X_{r,i}^g + S_r^g$$
**CGE Equation (5): Cobb-Douglas representation of demand of household, investment**
**and government**
$$DEM_r^d = A_r^d \cdot \prod_{i=1}^{N}(X_{r,i}^d)^{\alpha_{r,i}^d}$$





The first-order conditions for the optimality of the above problem imply the following
demand functions for household, government and investment, respectively:
**CGE Equation (6): demand function for household**
$$X_{r,i}^p = \frac{\alpha_r^p}{p_{r,i}} \cdot \left( \sum_{v=1}^V (\omega_{r,v} \cdot V_{r,v}) + \sum_j (pld_r \cdot QLAND_{r,j}) + \sum_{s,j} (p_{r,s}^{res} \cdot QRES_{r,j,s}) - T_r^d - S_r^p \right)$$

**CGE Equation (7): demand function for government**
$$X_{r,i}^g = \frac{\alpha_r^g}{p_{r,i}} \cdot \left( T_r^d + \sum_j T_{r,j}^z + \sum_j T_{r,j}^m - S_r^g \right)$$

**CGE Equation (8): demand function for investment**
$$X_{r,i}^n = \frac{\alpha_r^n}{p_{r,i}} \cdot \left( S_r^p + S_r^g + \varepsilon \cdot S_r^f \right)$$

Where
$DEM_r^d$ is final demand of households - p, investment - n and government - g;
$\omega_{r,v}$ is price of the $v^{th}$ primary factor;
$V_{r,v}$ is $v^{th}$ primary factor endowment by household;
$pld_r$ is land price;
$QLAND_{r,j}$ is land in sector $j$;
$p_{r,s}^{res}$ is price of resource $s$;
$QRES_{r,j,s}$ is quantity of resource $s$ in sector $j$;
$T_r^d$ is direct tax;
$S_r^p$ is household savings;
$T_{r,j}^z$ is production tax in sector $j$;
$T_{r,j}^m$ is import tariff of commodity $j$;
$S_r^g$ is government savings;
$S_r^f$ is current account deficits in foreign currency terms (or alternatively foreign savings);
$\varepsilon$ is foreign exchange rate;
$p_{r,i}$ is commodity price;
$X_{r,i}^d$ is final consumption of commodity $i$ by agent $d$ ($d \in$ households - p, investment - n and
government - g).
$A_r^d$ is the scaling parameter in Cobb-Douglas function by agent $d$ ($d \in$ households - p, investment
- n and government - g);
$\alpha_{r,i}^d$ is the share parameter in Cobb-Douglas function by agent $d$ ($d \in$ households - p, investment
- n and government - g);



Commodity supply and inter-regional trade

Supply of commodity adopts Armington assumption(Armington 1969), assuming that goods

produced from other provinces and abroad are imperfectly substitutable for domestically and
locally produced goods. This approach is shown in *Figure* and CGE Equation 9 and 10 below.

Supply to international region (f)

**CGE Equation (9): Armington representation of domestically produced and imported**
**commodity**
$$X_{fi} = CES_{s1}\{D_{ffi}, CES_{s2}(P_{1fi}, \cdots, P_{pfi})\}$$

Where

$D_{ffi}$ is commodity produced in the rest of world;

$P_{pfi}$ is commodity produced in China's provinces and exported to the rest of world;

•   Supply to China province (p)
**CGE Equation (10): representation of commodity produced locally and produced in**
**other provinces**
$$X_{pi} = CES_{s1}\{F_{fpi}, CES_{s2}(D_{ppi}, CES_{s3}(P_{1pi}, \cdots, P_{p'pi}))\}$$

Where

$F_{fpi}$ is commodity produced in the rest of world and imported by China's province;

$D_{ppi}$ is commodity produced in the province and supplied to the same province;

$P_{p'pi}$ is commodity produced in the other provinces.







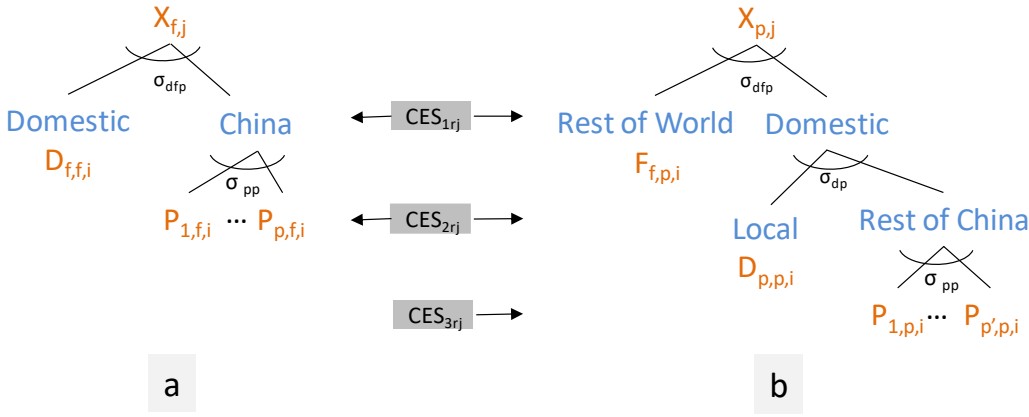

a

b



657 *Figure A2 Aggregation of local, domestic and foreign varieties of good*

658 *a, international regions; b, China provinces. σ is elasticity of substitution for inputs*


660 Two types of price variables are distinguished. One is prices in terms of the domestic currency

661 $p_i^e$ and $p_i^m$; the other is prices in terms of the foreign currency $p_i^{We}$ and $p_i^{Wm}$. They are linked with

662 each other as follows:

663 **CGE Equation (11)**

$$p_i^e = \varepsilon \cdot p_i^{We}$$

665 **CGE Equation (12)**

$$p_i^m = \varepsilon \cdot p_i^{Wm}$$

667 Furthermore, it is assumed that the economy faces balance of payments constraints, which is

668 described with export and import prices in foreign currency terms:

669 **CGE Equation (13)**

$$\sum_i p_i^{We} \cdot E_{r,i} + S_r^f = \sum_i p_i^{Wm} \cdot M_i$$

671 Where



| 672 | $E_{r,i}$ | Export of i-th commodity in region r, |
| 673 | $M_{r,i}$ | Import of i-th commodity in region r, |
| 674 | $p_i^{We}$ | Export price in terms of foreign currency, |
| 675 | $p_i^e$ | Export price in terms of domestic currency, |
| 676 | $p_i^{Wm}$ | Import price in terms of foreign currency, |
| 677 | $p_i^m$ | Import price in terms of domestic currency. |


**679 Market clearance**

The market-clearing conditions hold for both commodity and factor markets.
For the commodity markets described in CGE Equation 14, output $Q_{ri}$ in the corresponding
sector j (i=j) is equal to the total demand of intermediate inputs, household, investment and
government ($\sum_d X_{r,i}^d$), plus export to other international regions ($\sum_f F_{r,f,i}$) and provinces
($\sum_p P_{r,p,i}$), minus import from other international regions $\sum_f F_{f,r,i}$ and provinces ($\sum_p P_{p,r,i}$), and
plus net stock change ($STK_{ri}$):

**686    CGE Equation (14): market clearance of commodity and services**

$$Q_{ri} = \sum_d X_{r,i}^d + \sum_f F_{r,f,i} + \sum_p P_{r,p,i} - \sum_f F_{f,r,i} - \sum_p P_{p,r,i} + STK_{r,i}$$
For the factor markets described in CGE Equation 15, supply of total factor ($V_{r,v}$) is equal to
factor inputs in all sectors ($v_{r,v,j}$):

**691    CGE Equation (15): market clearance of production factor**

$$V_{r,v} = \sum_j v_{r,v,j}$$

**693 Macro closure**



In a CGE model, the issue of macro closure is the choice of exogenous variables, including
macro closure of investment-saving balance and current account balance. In this CGE model,
government savings ($S_r^g$), total investment and balanced of payment are fixed exogenously, and
foreign exchange rate is an endogenous variable.
**Dynamic process**
The model is solved at one-year time step in a recursive dynamic manner, in which the
parameters of capital stock (CGE Equation 16 and 17), labor force (CGE Equation 18), land,
natural resource, efficiency (CGE Equation 19), and extraction cost of fossil fuels are updated
based on the modelling of inter-temporal behavior and results of previous periods.
•   Capital accumulation process:
**CGE Equation (16): total investment demand**
$$TI_{r,t+1} = \sum_j CAPSTK_{r,j,t} \cdot [(1 + g_{r,t+1})^T - (1 - d_r)^T]$$

**CGE Equation (17): capital accumulation process**
$$CAPSTK_{r,j,t} = (1 - d_r)^T \cdot CAPSTK_{r,j,t-1} + T \cdot I_{r,j,t-1}$$

Where total investment ($TI_{r,t}$) is given exogenously, investment in sector j in the previous
period ($I_{r,j,t-1}$) is determined by the model depending on the rate of return to capital, capital stock
accumulation ($CAPSTK_{r,j,t}$) follows CGE Equation 18, $d_r$ is the depreciation rate (5% for all
regions), and T is time step (1 year).

Supply of total labor, land and resource:
**CGE Equation (18): factor growth pattern**
$$V_{r,v}^t = V_{r,v}^{t-1} \cdot (1 + gr_{r,v}^t)$$

Where $V_{r,v}^t$ is primary factor (v) of labor force, land and resource, and $gr_{r,v}^t$ is the
corresponding exogenous growth rate.
Efficiency parameters:



The CGE model distinguishes technological efficiency improvement of new investments

from that of existing capital stock.

For new investments, sectoral efficiencies of energy, land productivity and total factor

productivity are given as exogenous scenarios, while for existing capital stock, efficiency of par

(par efficiency of energy and capital) in time t ($EFF_{r,par,j}^{ext,t}$) is the average of capital stock

($EFF_{r,par,j}^{ext,t-1}$) and new investments ($EFF_{r,par,j}^{new,t-1}$) in the previous period, as per CGE Equation 19

here:

**CGE Equation (19): updating of efficiency parameters**

$$EFF_{r,par,j}^{ext,t} = \frac{(EFF_{r,par,j}^{ext,t-1} \cdot CAPSTK_{r,j,t-1} + EFF_{r,par,j}^{new,t-1} \cdot I_{r,j,t-1}) \cdot (1-d_r)^T}{CAPSTK_{r,j,t}}$$

**Data**

Most of the global data in the CGE model are based on GTAP 6 Dimaranan (Dimaranan and

730       V. 2006) and IEA(IEA 2009). China-specific provincial data sources are the 2002 inter-regional

input-output tables (IOT)(Li 2010) and the 2002 energy balance tables (EBT)(National Bureau of

Statistics of China (NBS) 2003). In addition, carbon emission factors; energy prices for coal, oil

and gas(National Bureau of Statistics of China (NBS) 2013) and renewable energy technology

costs (China National Renewable Energy Centre(China National Renewable Energy Centre

(CNREC) 2014) are also required. All the datasets are currently converted to the base year of 2002.

Moreover, it is well known that IOT and EBT are inconsistent when it comes to energy

consumption across sectors, and the energy data from EBT is regarded as more reliable than IOT.

A novel characteristic of this CGE model is that the IOT of China is consistent with the sectoral

energy consumption from China's EBT. To achieve this consistency, this study used the linear least

square method, as described in CGE Equation 20 - 23 below.

Minimizing:

**CGE Equation (20):**

$$\varepsilon = \sum_{en,j}(Shr_{en,j}^{IOT} - Shr_{en,j}^{EBT})^2$$

Subject to:



**CGE Equation (21):**

$$Shr_{en,j}^{IOT} = \frac{EN_{en,j}^{IOT}}{TCON_{en}^{IOT}}$$

**CGE Equation (22):**

$$Shr_{en,j}^{EBT} = \frac{EN_{en,j}^{EBT})}{TCON_{en}^{EBT}}$$

**CGE Equation (23):**

$$\sum_j EN_{en,j}^{IOT} = \sum_j EN_{en,j}^{EBT} \cdot P_{en}$$

Where
$\varepsilon$: Error to be minimized
*en*: Energy commodities (coal, gas, oil, electricity)
*j*: Sector classification in *Table A2*.
$Shr_{en,j}^{IOT}$: Share of energy consumption across sectors in IOT (%)
$Shr_{en,j}^{EBT}$: Share of energy consumption across sectors in EBT (%) according to (National
Bureau of Statistics of China (NBS) 2008)
$EN_{en,j}^{IOT}$: Energy consumption of *en* in sector *j* in IOT (USD)
$EN_{en,j}^{EBT}$: Energy consumption of *en* in sector *j* in EBT (PJ)
$TCON_{en}^{IOT}$: Total energy consumption of *en* in IOT (USD)
$TCON_{en}^{EBT}$: Total energy consumption of *en* in EBT (PJ)
$P_{en}$: Price of energy *en* (USD/PJ)

## 763    6.3 GEOS-Chem model

**General introduction**



The Chinese air quality is simulated in the global chemical transport model, GEOS-Chem
version v10-01(Atmospheric transport and chemistry model)[1] . Simulations are performed at 1/2
degree latitude by 2/3 degree longitude horizontal resolution over China region embedded in a 4
degree latitude by 5 degree longitude global simulation[2]. The model is driven by the
meteorological data from the Goddard Earth Observing System (GEOS, version 5) of the NASA
Global Modeling Assimilation Office (GMAO). The model contains 47 vertical layers up to 0.01
hPa. GEOS-Chem uses the same advection algorithm with the GEOS general circulation model[3].
Convective transport in GEOS-Chem is computed from the convective mass fluxes in the
meteorological archive. Boundary layer mixing in GEOS-Chem is calculated by a non-local
scheme[4] . The wet deposition by rain is considered for both water-soluble aerosols and gases,
and the scavenging by snow and cold/mixed precipitation is also considered for aerosol. Dry
deposition is calculated based on the resistance-in-series scheme for all the species with
gravitational settling for dust and coarse sea salt[5] .
**Model chemistry**

---

: http://acmg.seas.harvard.edu/geos/
: The global model provides initial and boundary conditions for the China domain.
: http://gmao.gsfc.nasa.gov/GEOS/
:The non-local scheme takes into account the large eddy transport under unstable boundary layer

condition, which is not well represented by a "local" scheme.

5: The resistance-in-series scheme considers the aerodynamic, boundary resistance and canopy

surface resistances during dry deposition process.



GEOS-Chem includes a detailed chemistry for 156 gas phase and aerosol phase species and
479 chemical reactions. The simulation contains a gas phase $HO_x$-$NO_x$-VOC-ozone-BrOx
chemistry, which considers the production and loss of ozone through reacting with $HO_x$, $NO_x$,
VOC and BrOx. GEOS-Chem also includes a detailed sulfate-nitrate-ammonium-carbonaceous-
dust-seasalt aerosol chemistry, which is coupled to gas phase chemistry. GEOS-Chem also
includes a detailed sulfate-nitrate-ammonium-carbonaceous-dust-seasalt aerosol chemistry, which
is coupled to gas phase chemistry. The ozone chemistry was first presented by Bey et al. with
additional oxidant-aerosol coupled simulation conducted by Park et al.(Park, et al. 2004). Recent
update includes the inclusion of bromine chemistry by Parrella et al. (Parrella, et al. 2012) .
Besides the anthropogenic emissions of ozone precursors as described in section X, we also
consider the contribution of natural sources. Biofuel emissions are based on Yevich and Logan's
study (Yevich and Logan 2003). Biogenic VOC emissions follow the MEGAN inventory
(Guenther 2006). The GFED version 3 is used to characterize biomass burning
emissions(Randerson, et al. 2007). We consider the lighting NO emissions based on the scheme
of Price and Rind with vertical distribution following Pickering et al.(Pickering, et al. 1998). For
the stratosphere, we use the linearized stratospheric O3 chemistry scheme(McLinden, et al. 2000).
About the methane, we assume the same methane concentrations as the other scenarios(Zhang, et
al. 2011).
## 6.4 GAINS model
**Model description**
GAINS model was developed by the International Institute for Applied Systems Analysis
(IIASA) in Austria, originally as the Regional Air Pollution Information and Simulation (RAINS)



model to estimate air pollutant emissions and design abatement strategies in Europe. It provides a
consistent framework for estimating emissions, mitigation potentials, and costs for air pollutants
($SO_2$, $NO_x$, PM, $NH_3$, NMVOC) and greenhouse gases (GHGs) included in the Kyoto protocol[26–
28]. GAINS-China is an application of the GAINS model for East Asia. Documentation on the
model and access to principal data, assumptions, and results are freely available online. Various
air-pollutant-mitigation technologies were considered in GAINS-China model. However, energy
and climate policies targeting carbon dioxide emissions were reflected implicitly through
alternative exogenous scenarios. The GAINS-China model provides annual air pollutant emissions
and pollution control costs data for China.

The basic principles of calculating emissions and emission control costs in the model present

in Equation 9 and 10. Components appearing on the right side of the equations are organized into
three different data categories: activity pathways, emission vectors, and control strategies. Each
emission scenario in GAINS is created through a combination of these three data categories.
Emissions-generating economic Activities are organized into activity pathways which are divided
into five groups: Agriculture (AGR), Energy (ENE), Mobile (MOB), Process (PROC), and VOC
sources (VOCP). This study mainly focuses on Energy and Mobile sources activity.
*Table A4 Mitigation technologies adapted in this paper*

| Air pollutant | | Abbreviation | Name | Application sector |
|---|---|---|---|---|
| $SO_2$ | 1 | LINJ | Limestone injection | Industry, power plants |
| | 2 | WFGD | Wet flue gas desulfurization | Industry, power plants |
| NOx | 1 | PHCCM | Combustion modification on existing hard coal power plants | Power plants |
| | 2 | HDSE | Stage control on heavy duty vehicles with spark ignition engines | Transport |
| | 3 | CAGEU | Stage control on construction and agriculture mobile sources | Transport |





| | | | | |
|---|---|---|---|---|
| | 4 | HDEU(I-VI) | EURO I-VI on heavy duty diesel road vehicles | Transport |
| | 5 | MMO2(1-3) | Stage control on motorcycles and mopeds (2-stroke engines) | Transport |
| | 6 | LFEU(I-VI) | EURO I-VI on light duty spark ignition road vehicles (4-stroke engines) | Transport |
| | 7 | MDEU(I-VI) | EURO I-VI on light duty diesel road vehicles | Transport |
| PM | 1 | CYC | Cyclone | Industry, power plants |
| | 2 | ESP | Electrostatic precipitator | Industry, power plants, industrial process |
| | 3 | HED | High efficiency deduster | Industrial process |
| | 4 | GP | Good practice | Industrial process |
| VOC | 1 | HDSE | Stage control on heavy duty vehicles with spark ignition engines | Transport |
| | 2 | CAGEU | Stage control on construction and agriculture mobile sources | Transport |
| | 3 | HDEU(I-VI) | EURO I-VI on heavy duty diesel road vehicles | Transport |
| | 4 | MMO2(1-3) | Stage control on motorcycles and mopeds (2-stroke engines) | Transport |
| CH4 | 1 | BC_DEGAS | Brown Coal pre-mining degasification | Industry, power plants, industrial process |
| | 2 | CH4_REC | Upgraded mine gas recovery and utilization | Industry, power plants, industrial process |
| | 3 | CH4_USE | Mining gas recovery and utilization of gas for energy purposes | Industry, power plants, industrial process |
| | 4 | FP_IMP | Fireplace improved | power plants, industrial process |
| | 5 | FP_NEW | Fireplace new | power plants, industrial process |


**GAINS Equation (1):**





$$Emissions = \sum_{i,t} A\,ctivity_i \times F_{t,i} \times (1 - R_{t,i}) \times C_{t,i}$$

**GAINS Equation (2):**
$$Costs = \sum_{i,t} A\,ctivity_i \times U_{t,i} \times C_{t,i}$$

Where
$F_{t,i}$:: emission factors of activities,
$R_{t,i}$: removal efficiencies of control technology t in activity i,
$U_{t,i}$: unit cost of control technology t in activity i, together with all background information,
form the so-called emission vectors.
$C_{t,i}$: control technology for each activity specified in control strategies.
•

Conversion tables are developed to make the database of the CGE and GAINS models match each
other. There are two types of conversion tables, namely conversion table for sector integration and
for fuel type integration. Each type of conversion tables are given in *Table A5* by taking Beijing
2005 as an example.





• *Table A5 Conversion table for sector match (BEIJ 2005 as an example).*

| GAINS Sectors | 2005 Activity by GAINS-China [PJ] | powerplant | DOM | IN_CHEM | IN_CON | IN_PAP | IN_IS_NFME | IN_NMMI | IN_OTH | TRA |
|---|---|---|---|---|---|---|---|---|---|---|
| PP_EX_WB | 0.0 | 0% | | | | | | | | |
| PP_EX_OTH | 23.5 | 23% | | | | | | | | |
| PP_EX_L | 0.0 | 0% | | | | | | | | |
| PP_EX_S | 15.3 | 15% | | | | | | | | |
| PP_NEW | 2.1 | 2% | | | | | | | | |
| PP_NEW_CCS | 2.1 | 2% | | | | | | | | |
| PP_NEW_L | 58.8 | 58% | | | | | | | | |
| PP_MOD | 0.0 | 0% | | | | | | | | |
| PP_MOD_CCS | 0.0 | 0% | | | | | | | | |
| PP_IGCC | 0.0 | 0% | | | | | | | | |
| PP_IGCC_CCS | 0.0 | 0% | | | | | | | | |
| PP_ENG | 0.0 | 0% | | | | | | | | |
| CON_COMB | 54.2 | | | | | | | | | |
| CON_LOSS | 46.2 | | | | | | | | | |
| IN_BO_CHEM | 0.0 | | | 50% | | | | | | |
| IN_BO_CON | 0.0 | | | | 100% | | | | | |
| IN_BO_OTH | 139.6 | | | | | | | | 73% | |
| IN_BO_OTH_L | 4.5 | | | | | | | | 2% | |
| IN_BO_OTH_S | 45.8 | | | | | | | | 24% | |
| IN_BO_PAP | 0.0 | | | | | 50% | | | | |
| IN_OC_ISTE | 0.0 | | | | | | 50% | | | |
| IN_OC_CHEM | 0.0 | | | 50% | | | | | | |
| IN_OC_NFME | 0.0 | | | | | | 50% | | | |
| IN_OC_NMMI | 0.0 | | | | | | | 100% | | |
| IN_OC_PAP | 0.0 | | | | | 50% | | | | |
| IN_OC_OTH | 0.0 | | | | | | | | 0% | |
| NONEN | 238.9 | | | | | | | | | |
| DOM | 399.2 | | 100% | | | | | | | |
| TRA_OT | 0.0 | | | | | | | | | 0% |
| TRA_OTS_L | 16.2 | | | | | | | | | 5% |
| TRA_OTS_M | 16.2 | | | | | | | | | 5% |
| TRA_OT_AGR | 7.3 | | | | | | | | | 2% |
| TRA_OT_AIR | 0.0 | | | | | | | | | 0% |
| TRA_OT_CNS | 11.0 | | | | | | | | | 4% |
| TRA_OT_INW | 0.0 | | | | | | | | | 0% |
| TRA_OT_LB | 0.0 | | | | | | | | | 0% |
| TRA_OT_LD2 | 0.0 | | | | | | | | | 0% |
| TRA_OT_RAI | 16.2 | | | | | | | | | 5% |
| TRA_RD | 0.0 | | | | | | | | | 0% |
| TRA_RD_HDB | 20.7 | | | | | | | | | 7% |
| TRA_RD_HDT | 76.2 | | | | | | | | | 25% |
| TRA_RD_LD2 | 0.2 | | | | | | | | | 0% |
| TRA_RD_LD4C | 95.6 | | | | | | | | | 32% |
| TRA_RD_LD4T | 38.1 | | | | | | | | | 13% |
| TRA_RD_M4 | 1.4 | | | | | | | | | 0% |

•

**Air pollution control technology penetration rate**

The technology penetration rates are given according to sectors, fuel types, regions and air
pollutants ($SO_2$, NOx, PM). Mitigation technology and penetration rate are different in different
sectors, process and provinces. GAINS-China model can provide very detailed data for 30
provinces. Hence, it is difficult to list all the penetration rates. *Table A* shows the $SO_2$ mitigation
technology penetration rate in different scenario years by taking Beijing as an example.



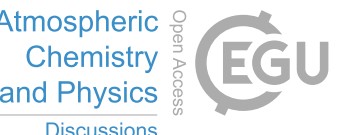

*Table A6 SO$_2$ mitigation technology and penetration rate (%) in Beijing under wPol scenario.*

| Sector-Fuel-Technology | Region | 2005 | 2010 | 2015 | 2020 | 2030 |
|---|---|---|---|---|---|---|
| CON_COMB-HC1-IWFGD | BEIJ | 9 | 15 | 15 | 15 | 15 |
| CON_COMB-HC1-LINJ | BEIJ | 10 | 14 | 24 | 34 | 53 |
| CON_COMB-HC2-IWFGD | BEIJ | 9 | 15 | 15 | 15 | 15 |
| CON_COMB-HC2-LINJ | BEIJ | 10 | 14 | 24 | 34 | 53 |
| IN_BO_OTH_S-HC1-IWFGD | BEIJ | 9 | 15 | 15 | 15 | 15 |
| IN_BO_OTH_S-HC1-LINJ | BEIJ | 10 | 14 | 24 | 34 | 53 |
| IN_BO_OTH_S-HC2-IWFGD | BEIJ | 9 | 15 | 15 | 15 | 15 |
| IN_BO_OTH_S-HC2-LINJ | BEIJ | 10 | 14 | 24 | 34 | 53 |
| IN_BO_OTH_S-HC3-IWFGD | BEIJ | 16.25 | 25 | 30 | 35 | 50 |
| IN_BO_OTH_S-HC3-LINJ | BEIJ | 10 | 10 | 10 | 10 | 10 |
| IN_OC-HC1-IWFGD | BEIJ | 9 | 15 | 15 | 15 | 15 |
| IN_OC-HC1-LINJ | BEIJ | 10 | 14 | 24 | 34 | 53 |
| IN_OC-HC2-IWFGD | BEIJ | 9 | 15 | 15 | 15 | 15 |
| IN_OC-HC2-LINJ | BEIJ | 10 | 14 | 24 | 34 | 53 |
| IN_OC-HC3-IWFGD | BEIJ | 16.25 | 25 | 30 | 35 | 50 |
| IN_OC-HC3-LINJ | BEIJ | 10 | 10 | 10 | 10 | 10 |
| PP_EX_L-HC1-LINJ | BEIJ | 0 | 0 | 40 | 40 | 40 |
| PP_EX_L-HC1-PRWFGD | BEIJ | 18 | 60 | 60 | 60 | 60 |
| PP_EX_L-HC2-LINJ | BEIJ | 0 | 0 | 40 | 40 | 40 |
| PP_EX_L-HC2-PRWFGD | BEIJ | 18 | 60 | 60 | 60 | 60 |
| PP_EX_L-HC3-LINJ | BEIJ | 0 | 0 | 20 | 30 | 40 |
| PP_EX_L-HC3-PWFGD | BEIJ | 38.89 | 60 | 60 | 60 | 60 |
| PP_EX_S-HC1-LINJ | BEIJ | 0 | 0 | 40 | 40 | 40 |
| PP_EX_S-HC1-PRWFGD | BEIJ | 18 | 60 | 60 | 60 | 60 |
| PP_EX_S-HC2-LINJ | BEIJ | 0 | 0 | 40 | 40 | 40 |
| PP_EX_S-HC2-PRWFGD | BEIJ | 18 | 60 | 60 | 60 | 60 |
| PP_EX_S-HC3-LINJ | BEIJ | 0 | 0 | 20 | 30 | 40 |
| PP_EX_S-HC3-PWFGD | BEIJ | 38.89 | 60 | 60 | 60 | 60 |
| PP_MOD-HC1-LINJ | BEIJ | 0 | 0 | 30 | 30 | 30 |
| PP_MOD-HC1-PWFGD | BEIJ | 18 | 60 | 70 | 70 | 70 |
| PP_MOD-HC2-LINJ | BEIJ | 0 | 0 | 30 | 30 | 30 |

| | | | | | | |
|---|---|---|---|---|---|---|
| PP_MOD-HC2-PWFGD | BEIJ | 18 | 60 | 70 | 70 | 70 |
| PP_MOD-HC3-LINJ | BEIJ | 0 | 0 | 25 | 25 | 30 |
| PP_MOD-HC3-PWFGD | BEIJ | 38.89 | 60 | 65 | 70 | 70 |
| PP_NEW_L-HC1-LINJ | BEIJ | 0 | 0 | 30 | 30 | 30 |
| PP_NEW_L-HC1-PWFGD | BEIJ | 18 | 60 | 70 | 70 | 70 |
| PP_NEW_L-HC2-LINJ | BEIJ | 0 | 0 | 30 | 30 | 30 |
| PP_NEW_L-HC2-PWFGD | BEIJ | 18 | 60 | 70 | 70 | 70 |
| PP_NEW_L-HC3-LINJ | BEIJ | 0 | 0 | 25 | 25 | 30 |
| PP_NEW_L-HC3-PWFGD | BEIJ | 38.89 | 60 | 65 | 70 | 70 |

Note: HC1, HC2 and HC3 represent hard coal grade 1, grade 2 and grade 3, respectively. CON_COMB
represents other energy sector-combustion. IN_BO_OTH_S, IN_OC represent Industry: other sectors;
combustion of brown coal/lignite and hard coal in small boilers (<20 MWth) and Industry: other combustion,
respectively. PP_EX_L, PP_EX_S, PP_MOD and PP_NEW_L represent Exist large scale power plants,
Exist small scale power plants, Modern power plants (supercritical, ultra-supercritical) and New large scale
power plants, respectively.

## 6.5 Additional results

**Energy consumption**

AIM/CGE-China model provides energy consumption data of 30 provinces to GAINS-China

model.





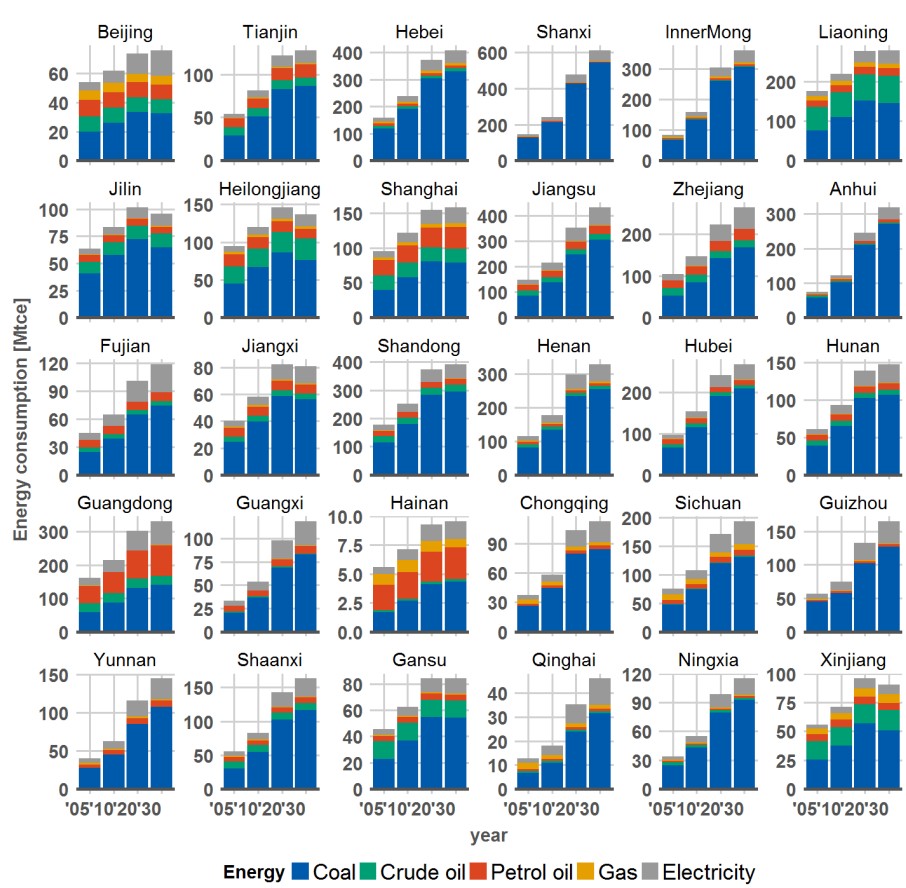


*FigureA3 Energy consumption from 2005 to 2030 in 30 provinces in China.*
**Primary emissions**





Figure A4: National primary emissions [a] and regional average concentration of ozone and $PM_{2.5}$ [b].




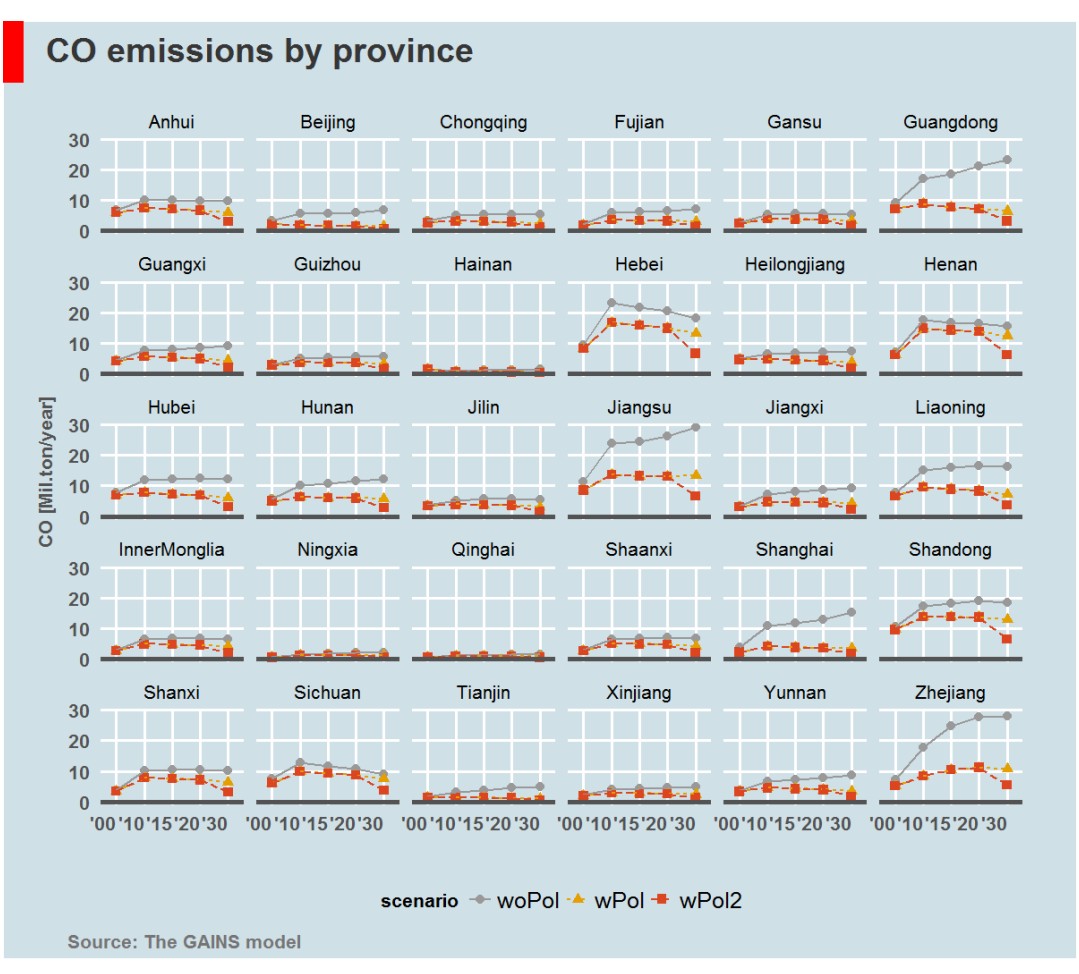


Figure A5: Provincial primary emissions of CO.



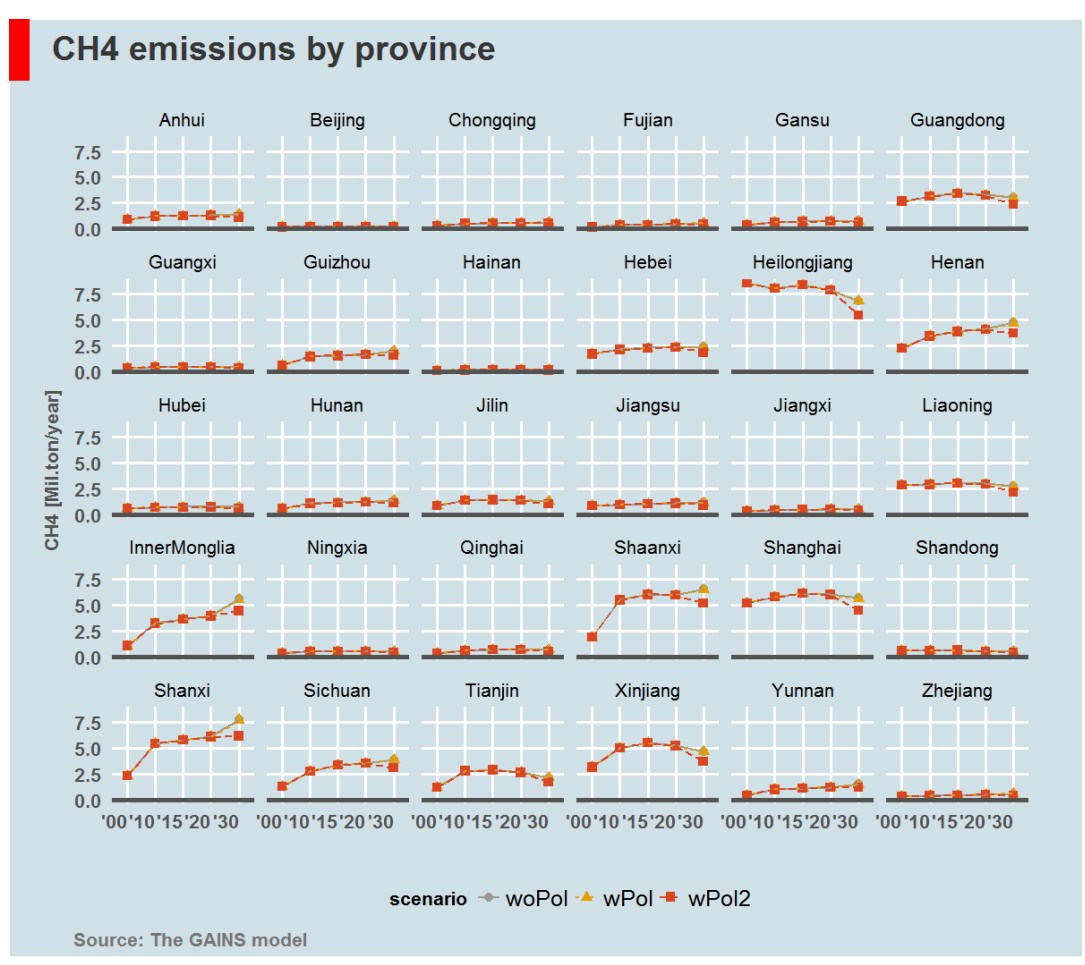


Figure A6: Provincial primary emissions of CH₄.




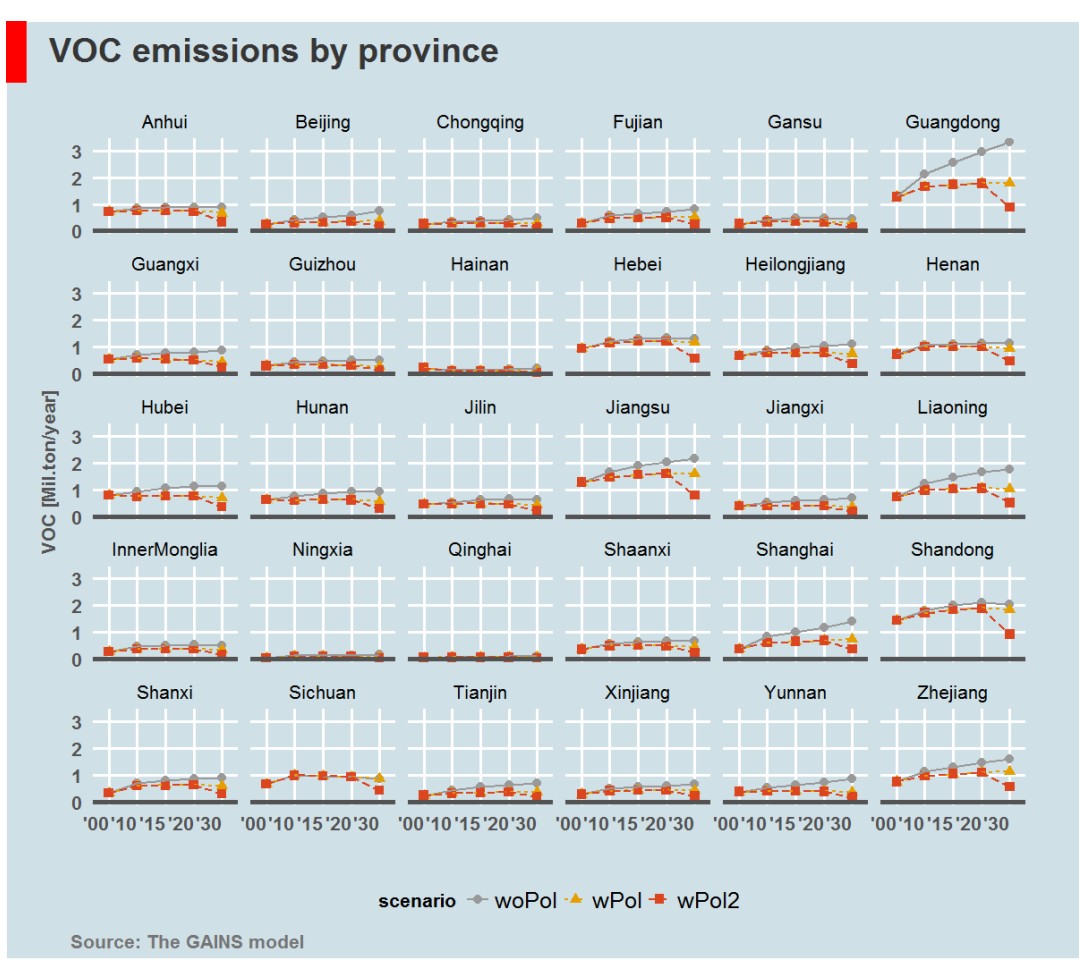


Figure A7: Provincial primary emissions of VOC.




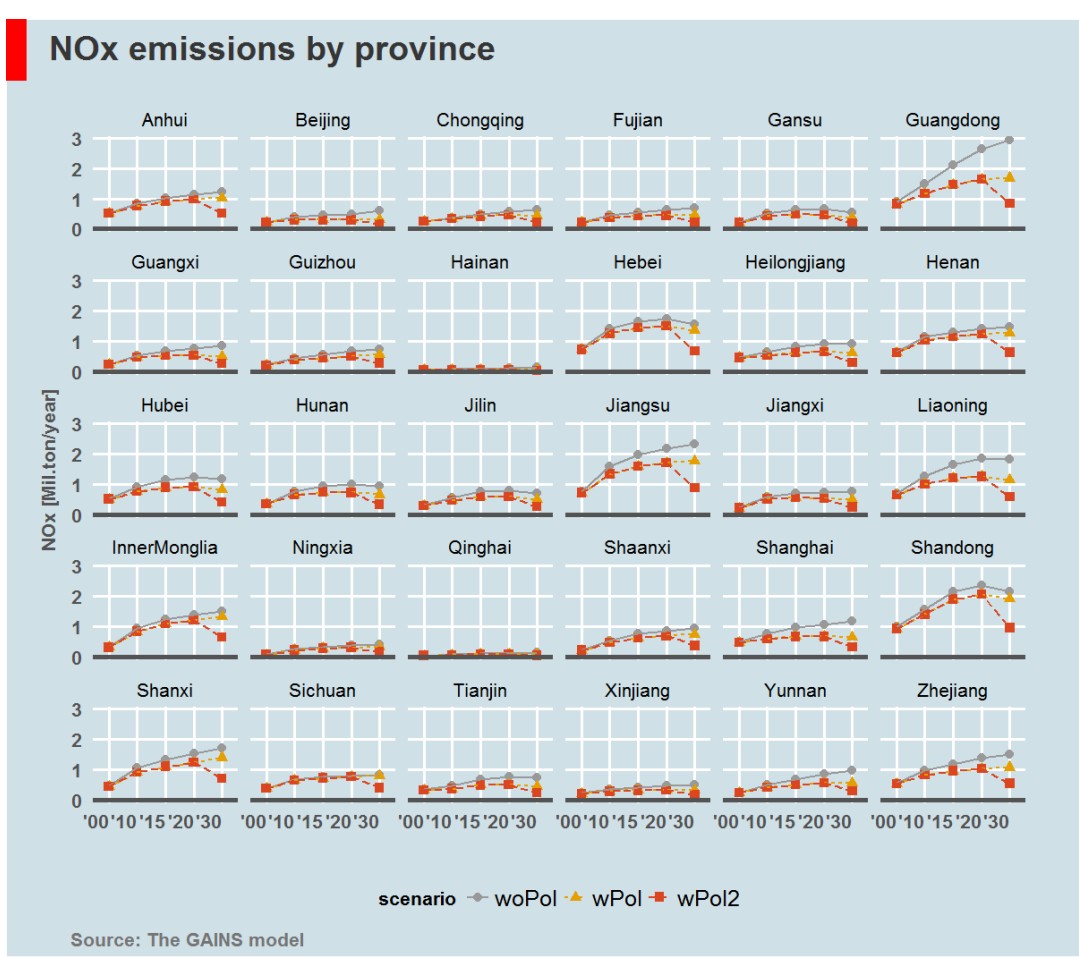

Figure A8: Provincial primary emissions of $NO_x$.

**Seasonal average concentration**



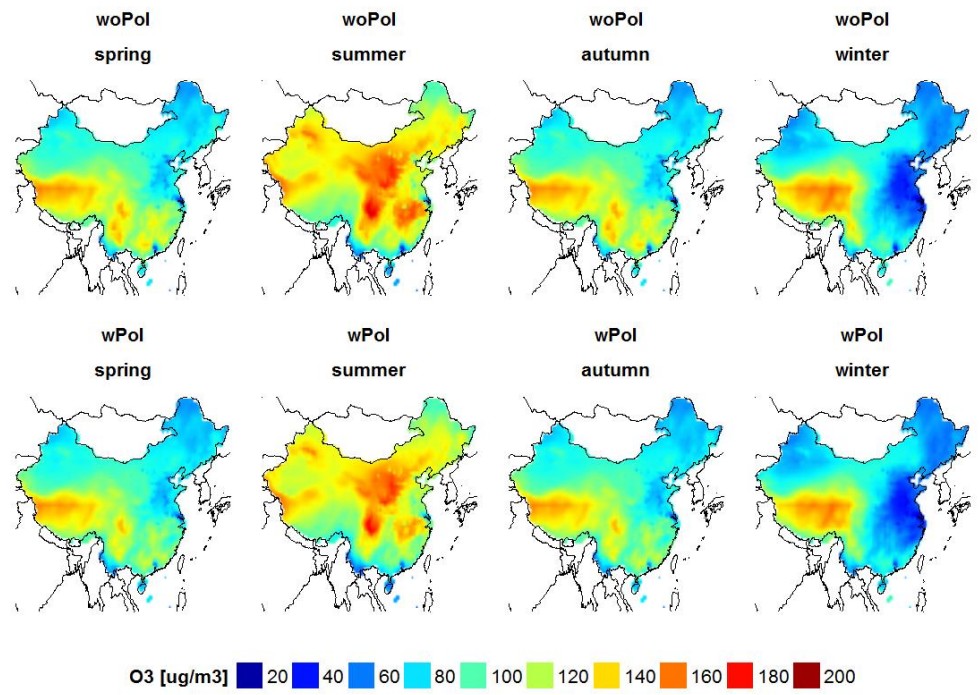


Figure A9: Seasonal variation of daily maximum 8-hour mean concentration of ozone in 2030.



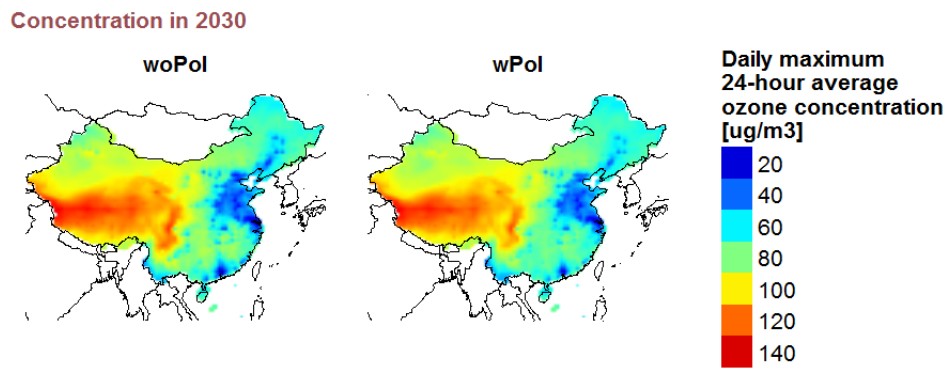

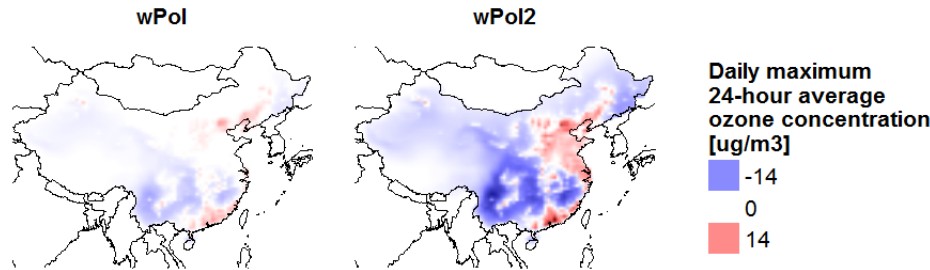

Figure A10: Daily maximum 24-hour ozone concentration in woPol and wPol scenarios (upper)
and change from woPol to wPol and wPol2 scenarios (lower).



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
