# Peer review of "Health and Economic Impacts of Ozone Pollution in China: a provincial level analysis"

_Atmospheric Chemistry and Physics, 2017_

## Referee Comment (RC1) · Anonymous Referee #1 · 30 Oct 2017

This paper combines different types of models to estimate the current burden of ozone on public health and economic impacts in Chinese provinces, as well as the benefits of more stringent policies in the future. Overall, the study is interesting, adds value to the literature since most similar papers have focused on PM2.5 in China and not ozone, and the broad and multi-disciplinary range of impacts addressed is impressive. The general approach has been used by many other studies to estimate future air pollution-related health and economic impacts of emission policies, and this paper goes farther with the CGE model integration. Each model applied is generally established and widely used. I focused my review on the air quality modeling and health impact assessment.

Major issues:

[Figure]

1. The health impact assessment methods were confusing and unclear. More detail and rationale behind several of the data inputs could improve this part of the paper quite a bit. There could also be some inappropriate assumptions or aspects of the methods, though it is hard to tell without clear descriptions of the methods. Specific comments on the health impacts aspect:

1a. It was very confusing whether and where PM2.5 impacts were included. The paper focuses on ozone, the results section appears to be ozone health impacts only, but the conclusions anad abstract compare ozone health impacts to PM2.5 impacts. The methods section in the Supplemental material jumble ozone and PM2.5 together. Please clarify whether PM2.5 health impacts used to compare against the ozone impacts were original calculations done by the authors, or drawn from somewhere else. Are the PM2.5 health impacts included in the numbers given in the results and abstract sections?

1b. Line 93: Was SSP2 used to project future population, but not baseline disease rates? Please clarify, and if so, explain why. What was the source of baseline disease rates? What year are they from? Were they province-specific or national? Were they age specific? These choices can have really substantial influence on results and, in the case of this paper, could affect conclusions about the province-level results/rankings. Disease rates differ dramatically throughout China, so province-level rates should be used if possible. Also, both disease rates and age structure might change dramatically through 2030, with potentially substantial results.

1c. Clarify whether a log-linear or linear function was used. What is the evidence behind this choice?

1d. Is total mortality all-cause mortality or non-accidental mortality?

1e. What ages were calculated for each health endpoint? Where health impact calculations age specific? Was population age structure projected to the future?

1f. Discuss methods for calculating uncertainty, different sources of quantified and unquantified uncertainties – either in methods, health section, discussion, appendix, or preferably in at least two of these places. Line 384-385 is not sufficient.

1g. Table A1 – The sources list health impact assessments, sometimes for PM2.5 and not ozone, instead of the primary source of concentration=response functions. There should be a lot more information provided with this table, including the primary CRF source, the age it applies to, what ozone metric (e.g. 8-hr, 24-hr, seasonal avg, annual avg?), the concentration change it applies to, etc.

2 The air quality modeling portion of the study was also not entirely described, and some of the inadequately described assumptions and methods could have substantial influence on results. Specific questions on the air quality modeling methods:

2a. Does natural background include transport from other countries? What about wildfires and farmland, both of which may not be natural? The paper should discuss where the impacts from these sources are in the results, and how that might influence the province-level conclusions.

2b. Lines 101-102: How big of an impact would it have if meteorology was projected to 2030?

2c. Lines 101-102: About how many grid boxes per province? Are concentration results simple averages of the gridboxes within each province

Additional specific comments:

3. Abstract makes statements beyond the content and certainty presented in the paper. For ex. "will" on lines 27 and 29 might better reflect the amount of certainty this paper provides if written instead as "could be". Clarify in these lines whether these values are specific to ozone health impacts. Lines 32-34 seem to be beyond the scope of the paper.

4. Section 3.1 should summarize how much NOx and VOC emissions change by scenario, to help interpret some of the ozone concentration results in the coming sections. This information is currently in the Appendix figures, but should be summarized in the main text.

5. Lines 192-202: suggest providing some indication of how many of the provinces with the highest ozone levels fall into each of these three categories. And how many of them might be most affected by the portion of what is called natural, that is actually from international transport of air pollution, wildfires, and farmland. These may not be natural.

6. Line 201: "a lot"... by how much specifically?

7. Figure 3: shift x-axis labels to line up with tick points

8. Line 2014: "opportunities to obtain ozone-related diseases, premature death and payment..." is strange wording and should be revised

9. Line 235: 61.2% seems off. Is this calculated by dividing 140,100/491,000?

10. Lines 238 and 242: 4-5% and 1-2% of what? Clarify what these percentages represent.

11. Figure 4: How were different morbidity endpoints added together to get the results in the 2nd row? Is this equating all cases of different morbidity endpoints, so that the total number of cases is the cases of asthma + the cases of respiratory hospital admission + cases of cough, etc.?

12. Line 297: why does VSL change by scenario?

13. Line 342: Can forest burning and farmland really be considered natural?

14. Line 352: except in the cases where you found ozone increases...

15. Line 356-357: There are health effects below the WHO guideline. There is a large body of epidemiological literature on this. For example, see EPA PM National Ambient

Air Quality Standards Regulatory Impact Assessments. Suggest removing.

16. Line 362: Odd to introduce PM mortality results for the first time in the Discussion. Perhaps this should be woven into the health results section or left out completely since no methods information has been given to describe what was done to calculate that (e.g. shape of concentration-response function, epidemiological studies used, age ranges included)

17. Line 363-364: what about cardiovascular impacts?

18. Line 366: What does "2.3 times per capita per year" mean? Does this mean 2.3 cases per capita per year?

19. Line 376: "On the other hand" seems unnecessary since this line agrees with the last.

20. Figure 6: What is changing in the inputs underlying the results in this figure? Just concentrations? What about population, disease rates, etc.? Please clarify.

21. Figure 6: Clarify that the color legend at the bottom applies only to the bottom panel – e.g. by drawing a darker line between the 3rd and 4th rows. I was scratching my head about why there was so much asthma mortality in the top panels, because asthma is gray in the bottom panel.

22. Section 6.1: see major comment on health impact assessment above.

23. Lines 428-442 are perplexing. Many of these variables are not in the equations.

24. Line 426: Why is Ir all cause? Is this because only all-cause mortality for ozone exposure was calculated? If cause-specific mortality is calculated, the baseline disease rate should be cause-specific too.

25. Lines 445-449: So is this work lost due to death, or morbidity? This whole section is vague and confusing and should be revised.

26. Line 448: Only 4% of chronic mortality is aged between 15-65 years? That seems really low. Can provide some data to back this up? Need a source.

27. Line 784 is repetitive

28. Line 788: missing section reference
* * *

---

## Referee Comment (RC2) · Anonymous Referee #2 · 22 Nov 2017

This paper addresses an important topic - health and economic impacts of ozone pollution in China - with a set of state-of-the-art models. This type of analysis would be very doable given the models selected, but I have significant concerns about the choices made in this particular study, and the descriptions here lack enough detail to fully evaluate the outcomes. The information provided, however, suggests that there are some serious limitations in how the analysis was conducted.

On the health impact analysis, it is unclear which functions are used, whether analysis was done for ozone as well as PM. In particular, the literature lacks the most recent citations. For example, the recent results of Turner et al. (2016) suggest a larger impact of long-term ozone on mortality relative to PM2.5 than the functions used int his study. If the authors wish to make a point about the relative impacts, they should at least

discuss the implications of different choices of exposure-response functions.

With respect to the GEOS-Chem simulations, there is a lack of detail in explaining the simulations. Though GEOS-Chem is a well known model, the authors should provide some information on whether this particular version can capture the chemistry of ozone in China (whether this be through their own work or through citations to the same version of GEOS-Chem). Basic information is missing, for example the meteorological data. I assume that this is a nested-grid simulation, given the resolution and version of GEOS-Chem used; the description lacks the appropriate citations for this as well as details on how boundary conditions were used, as there are several different versions in the literature.

The economics results are described in ways that do not make sense with economic intuition. In particular, for the ACP audience, it would be useful to discuss the basis for using CGE vs. VSL. It is unclear whether the authors intend to use these together, and how they can justify this choice given the very different economic assumptions made in these two different approaches.

Overall, while the paper addresses an important topic with well-regarded models, the implementation has some major issues.

Specific comments:

Abstract, line 27-29: I assume that these numbers are talking about ozone only damages, but the result implies that analysis was conducted for PM2.5 as well. This should be clarified.

Introduction: It would be useful here to review other studies of air pollution which use CGE methods, and the pros and cons of using such approaches, relevant to this particular study.

Line 95-96: What previous study is being referred to here?

Table A1: I am confused by Table A1 here. It refers to concentration-response functions

[Figure]

from ozone related health impacts in the title, but the sources (e.g. Apte et al., 2015) refer to PM2.5. This could be clarified. I would suggest that the particular studies be credited in an additional column of the table, and the specific endpoints (O3 or PM) be identified.

Line 109-110: This is not a correct summary of the reference cited. In fact, Berman et al. (2012) note that the US EPA science assessment identify no threshold for the relationship between ozone exposure and premature mortality.

Line 110-112: It would be useful to recap the methods here. In particular, the methods for PM2.5 should be different from ozone, given the differences in types of outcomes. It is not clear how mortalities are covered in the CGE application. It does not really make economic sense for the morbidity to be evaluated using CGE and mortalities using VSL.

Line 187: Is this actually showing a realistic result, given the nonlinearities in ozone formation? Also, I'm not sure that the "natural background" is accurate as this level of ozone is not 'natural' in the sense of non-anthropogenic - non-China background would be a more accurate term.

---

## Author Comment (AC1) · 20 Jan 2018

To Editor and all reviewers Many thanks to the excellent comments raised by the reviewers on our paper. In this revision, we have the following important improvements: âǍć Improving clarity of presentation, including – Revising the additional results of PM2.5 and ozone. – Revising the abstract part. – Revising health assessment model only about ozone pollution. – Revising GEOS-Chem model about detailed simulation in 30 provinces – Clarifying the nature background and adding zero-out description – Updating the parameter ( in equation 3, meaning the share of mortality between 15 and 65 years old due to ambient air pollution, changed from 0.04 to 0.27 based on age-specific all-cause mortality from Global Burden of Disease Study 2010. âǍć Adding more results in appendix, including – Exposure-response functions for ozone pollution

and references

We hope the current revision is acceptable by the editor and reviewers. Please see the attachment for better formatted file.

Anonymous Referee #1

This paper combines different types of models to estimate the current burden of ozone on public health and economic impacts in Chinese provinces, as well as the benefits of more stringent policies in the future. Overall, the study is interesting, adds value to the literature since most similar papers have focused on PM2.5 in China and not ozone, and the broad and multi-disciplinary range of impacts addressed is impressive. The general approach has been used by many other studies to estimate future air pollution-related health and economic impacts of emission policies, and this paper goes farther with the CGE model integration. Each model applied is generally established and widely used. I focused my review on the air quality modeling and health impact assessment.

Response: Thank you very much for your positive comments on the interdisciplinary approach. In particular, we are more grateful for reading our manuscript so carefully and raising so many constructive major and minor comments that cover comprehensive aspects of this paper that really helped us to further improve our manuscript.

Major issues: 1. The health impact assessment methods were confusing and unclear. More detail and rationale behind several of the data inputs could improve this part of the paper quite a bit. There could also be some inappropriate assumptions or aspects of the methods, though it is hard to tell without clear descriptions of the methods. Specific comments on the health impacts aspect:

Response: Because of the length limitation of the manuscript, some detailed information is not described in the main text of manuscript, instead, we added more explanations about our health model in the Support Information. Further-

more, the up-to-date technical introduction to the CGE model and Health model is made publicly available at: http://scholar.pku.edu.cn/hanchengdai/imedcge and http://scholar.pku.edu.cn/hanchengdai/imedhel, respectively.

1a. It was very confusing whether and where PM2.5 impacts were included. The paper focuses on ozone, the results section appears to be ozone health impacts only, but the conclusions and abstract compare ozone health impacts to PM2.5 impacts. The methods section in the Supplemental material jumble ozone and PM2.5 together. Please clarify whether PM2.5 health impacts used to compare against the ozone impacts were original calculations done by the authors, or drawn from somewhere else. Are the PM2.5 health impacts included in the numbers given in the results and abstract sections?

Response: The current health model is extended from the previous version that only quantifies the health impacts of PM2.5 that was published in 2016 (Xie, Dai et al. 2016). However, in our previous paper, the PM2.5 concentration is calculated by the simple source-receptor matrix GAINS model, which is not as sophisticate as the GEOS-chem model used in this study. The current version covers both PM2.5 and Ozone health impacts with separate exposure-response functions for PM2.5 and ozone. All the health impacts showed in this manuscript is calculated simultaneously with one health model using the concentration results of PM2.5 and Ozone from the same round of simulation by GEOS-Chem model simulation, which strengthens the consistency of the comparison of health impacts between PM2.5 and ozone. Furthermore, we updated the parameter ( in equation 3, meaning the share of mortality between 14 and 65 years old due to ambient air pollution, changed from 0.04 to 0.27 based on based on age-specific all-cause mortality from Global Burden of Disease Study 2010 in the health model for both PM2.5 and ozone. Therefore, the results on PM2.5 health impacts in this paper is not a repetition of our 2016 paper, but is a more robust updated estimation. To make clear that PM2.5 result is indeed the part of the result of this paper, we have added an additional section of "3.4 Comparison of ozone and PM2.5 impact" in the result part

and moved figure 6 (Comparing health impacts between PM2.5 and ozone.) to this section. Moreover, we improved the whole storyline of the whole manuscript by adding one more research question of comparing PM2.5 and ozone health impacts.

1b. Line 93: Was SSP2 used to project future population, but not baseline disease rates? Please clarify, and if so, explain why. What was the source of baseline disease rates? What year are they from? Were they province-specific or national? Were they age-specific? These choices can have really substantial influence on results and, in the case of this paper, could affect conclusions about the province-level results/rankings. Disease rates differ dramatically throughout China, so province-level rates should be used if possible. Also, both disease rates and age structure might change dramatically through 2030, with potentially substantial results.

Response: We used population projection for 30 provinces in China from SSP2 but not for any future disease rates since there is not any project on disease rates in 30 provinces in China.

For natural mortality rate, since there is no data on age-specific natural death rate or future projection of death rate at the provincial level, we used the national average natural death rate projection from World Bank database for all age group and all simulation years (https://data.worldbank.org/data-catalog/population-projection-tables).

For the morbidity estimation, ERFs from (Bickel, Friedrich et al. 2005) are used for all the provinces in China. There is no such long-term study to show the relationship between ozone concentration and morbidity in 30 provinces in China. As shown in Table A1, the ERFs are different for different population groups, but not age-specific ERFs. We adopt the relationship between 1 ug/m3 incremental ozone concentration and the cases of each disease increase. We use the same relationships to estimate the total morbidity from ozone pollution in 30 provinces, but not province-specific ERFs. This will also lead to uncertainty in the estimation. However, because of the data unavailability, this is best available method for our estimation.

1c. Clarify whether a log-linear or linear function was used. What is the evidence behind this choice?

Response: For mortality from ozone pollution, we adopt the linear exposure-response function from(Turner, Jerrett et al. 2016). Jerrett et al. (2009) and Turner et al. 2016 provide the clear evidence for long-term impact of ozone pollution as a linear relationship. This study is a part of American Cancer Society(ACS) cohort study of U.S. adults over 30 years of age for 1977-2000. As you mentioned, there are quite few studies about health impact of ozone pollution, especially for long-term study. So we adopt the ERF from Turner et al. 2016. We modified in the health module in Support Information Ling 21 and 22 as "This study follows the methodology that the relative risk (RR) for the endpoint is in a linear relationship with the concentration level."

1d. Is total mortality all-cause mortality or non-accidental mortality?

Response: Total mortality we use in this study is all-cause mortality. We addressed the all-cause mortality in the health assessment module introduction Line 27 and Line 28 in support information as "The number of health endpoints is estimated by multiplying RR with population and reported all-cause mortality rate."

1e. What ages were calculated for each health endpoint? Where health impact calculations age specific? Was population age structure projected to the future?

Response: For mortality, we only calculate population over 30 years old. For morbidity, we cover all population by adopting different exposure response functions (Table A1 and Table A2 in Support Information). We use the age-structured population projection from SSP2 in the future with age structure(Samir and Lutz 2014). We added one sentence at Line 28 and 29 in Support Information as "This study uses the age-structured population projection from SSP2 in the future."

1f. Discuss methods for calculating uncertainty, different sources of quantified and unquantified uncertainties – either in methods, health section, discussion, appendix, or

preferably in at least two of these places. Line 384-385 is not sufficient.

Response: We use four different models and there are different uncertainties from different models and sources. For GAINS model, GEOS-Chem model and CGE model, we use the historic data to modify model to reduce uncertainties as many as possible. However, there are still a lot of uncertainty from this integrated approach. We quantified the uncertainty from exposure-response functions, which is most common uncertainty analysis for health impact assessment. Beside ERFs, the energy demand projection, population project and death rate project are also very important for the assessment. We add more discussion about uncertainty from Line 407 to Line 420 in our study as followed "Uncertainty within our framework could be classified into three sources. The first source is uncertainty of future economic development and energy consumption in the CGE model. The second source is estimation of future air pollutant emissions and ozone concentration, which is related to both technology selection and the behavior of the GEOS-Chem model. The last source is related to ERFs used in the health model. In terms of uncertainty of ERFs, the numbers in the parenthesis show 95% CI of ERFs. Besides these uncertainties, climate change also has impact on future ozone air quality and could have intersection effect on ozone precursor emissions. But in this study, we do not quantify the magnitude of the impact without a detailed model analysis. "

1g. Table A1 – The sources list health impact assessments, sometimes for PM2.5 and not ozone, instead of the primary source of concentration=response functions. There should be a lot more information provided with this table, including the primary CRF source, the age it applies to, what ozone metric (e.g. 8-hr, 24-hr, seasonal avg, annual avg?), the concentration change it applies to, etc.

Response: In our health assessment model, we aggregate the ERFs for both PM2.5 and ozone. To avoid confusion, we make one ERFs table for ozone (Table A1) and one table (Table A2) for PM2.5 in this paper. In this revision, we added more detailed information about ozone ERFs in the following table with 8-hr concentration per 1 ug/m3 change. We also add morbidity for different age groups.

Table A1 Exposure-Response Fuctions of ozone pollution Endpoint Population Impact category ER function C.I.95% low C.I.95% high Morbidity Entire age groups Respiratory hospital admissions<65 3.54E-06 6.12E-07 6.47E-06 Adults Respiratory hospital admissions 65+ 1.25E-05 -5.00E-06 3.00E-05 Adults Bronchodilator usage 1.04E-02 -3.64E-03 2.24E-02 Adults Lower respiratory symptoms 2.29E-03 -6.14E-03 1.16E-02 Adults Asthma 4.29E-03 3.30E-04 8.25E-03 Adults Minor restricted activity day 1.15E-02 4.40E-03 1.86E-02 Adults Consultation for allergic rhinitis 1.60E-04 1.22E-04 2.03E-04 Children Consultation for allergic rhinitis 3.03E-04 1.89E-04 4.29E-04 Children Cough 9.30E-02 -1.90E-02 2.22E-01 Children Lower respiratory symptoms(wheeze) 1.60E-02 -4.30E-02 8.10E-02 Children Consultation for allergic rhinitis 3.03E-04 1.89E-04 4.29E-04 Adults Consultation for allergic rhinitis 1.60E-04 1.22E-04 2.03E-04 Children Acute respiratory symptoms days 9.30E-02 -1.90E-02 2.22E-01 Adults Acute respiratory symptoms days 1.60E-02 -4.30E-02 8.10E-02 Mortality Adult 30+ Mortality from chronic exposure 2.00E-03 6.50E-04 3.35E-03 Work loss Adults Work loss day (days) 4.13E-03 1.65E-03 6.63E-03 VSL Entire age groups Value of statistical life (million USD) 2.50E-01 Source:(Bickel, Friedrich et al. 2005, Xie 2011, Turner, Jerrett et al. 2016)

Table A2 Concentration response functions for PM2.5-related health endpoints. Category Population Endpoint ERF C.I. (95%)Low C.I. (95%)High Morbidity Adult Work loss day 0.0207 0.0176 0.0238 All ages Respiratory hospital admissions 1.17E-05 6.38E-06 1.72E-05 Adult Cerebrovascular hospital admission 8.4E-06 6.47E-07 1.16E-05 Adult Cardiovascular hospital admissions 7.23E-06 3.62E-06 1.09E-05 Age 27+ Chronic bronchitis 4.42E-05 -1.8E-06 9.02E-05 All ages Asthma attacks 0.000122 4.33E-05 0.001208 All ages Respiratory symptoms days 0.025 0.217 0.405 Mortality Age 30+ All cause (international) 0.004 0.0003 0.008 Age 30+ All cause (China specific) 0.0009 -0.0003 0.0018 Adult RR_COPD Non-linear function Adult RR_LNC Adult RR_IHD_25y_65y Adult RR_STR_25y_65y Children under 5 RR_LRI Source(Pope III, Burnett et al. 2002, Bickel, Friedrich et al. 2005, Apte, Marshall et al. 2015)
2 The air quality modeling portion of the study was also not entirely described, and some of the inadequately described assumptions and methods could have substantial influence on results. Specific questions on the air quality modeling methods:

Response: We improve our manuscript and support information according to your suggestions. a) We added more description about GEOS-Chem model in Support Information and cited more references to support our result. b) We explained the "nature background" and change it to "zero-out" Line 409-412 in Support Information. c) We clarified resolution and how to aggregate provincial average concentration in our simulation Line 421-423 in Support Information.

2a. Does natural background include transport from other countries? What about wildfires and farmland, both of which may not be natural? The paper should discuss where the impacts from these sources are in the results, and how that might influence the province-level conclusions.

Response: The "natural background" simulation includes the transport from other countries contributed by both natural and anthropogenic sources. In this simulation, we zero-out all the Chinese anthropogenic emissions from industrial, energy, domestic, traffic, and agricultural sectors, but kept all the natural sources (e.g. wildfires and volcanos) unchanged. We did this because only this part of emissions is relevant for Chinese air pollution control policies, and this case represents an ideal (and extreme) scenario for emission controls. To avoid confusion, we modify the sentences in Line 208-214 as "To elucidate the contribution of natural and foreign sources to ozone formation, we conducted an additional simplified experiment in the GEOS-Chem model by reducing the Chinese anthropogenic emissions to zero and calculating the resulting daily maximum 8-hour ozone concentration. In this scenario, all the Chinese emissions from industrial, energy, domestic, traffic, and agricultural (only $NH_3$) sectors are assumed to be zero, and the modeled ozone concentrations are contributed by Chinese natural and foreign (both natural and anthropogenic) sources. This concentration is defined as "zero-out" in our case."

2b. Lines 101-102: How big of an impact would it have if meteorology was projected to 2030?

Response: We are glad that the referee mentioned this point. The change of climate would also have an impact on the future ozone air quality, and could have intersection effect on ozone precursor emissions. However, we cannot quantify the magnitude of the impact without a detailed model analysis. We added one sentence Line 111 and 112 in our manuscript as "In this study we do not considerate the meteorology impact in the future on ozone concentration, because global studies found climate change contributes lower impact on overall increase in ozone mortality than emissions".

2c. Lines 101-102: About how many grid boxes per province? Are concentration results simple averages of the grid boxes within each province Response: The size of provinces varies drastically in China. The number of grid boxes ranges from ∼600 in Xinjiang to ∼10 in Beijing. We used the simple arithmetic average of the ozone concentrations of all the grid boxes in a province for analysis.

Additional specific comments: 3. Abstract makes statements beyond the content and certainty presented in the paper. For ex. "will" on lines 27 and 29 might better reflect the amount of certainty this paper provides if written instead as "could be". Clarify in these lines whether these values are specific to ozone health impacts. Lines 32-34 seem to be beyond the scope of the paper.

Response: "Could be" is better than "will", because of the uncertainty in this integrated study. We changed the "will" to "could be". We also modified the words in other parts of this manuscript "Without a control policy, in 2030 China could experience a 78 billion CNY Gross Domestic Production (GDP) loss (equivalent to 0.09%), and a 2300 billion CNY (equivalent to 2.7% of GDP) life loss from ozone pollution. In contrast, with a control policy, the GDP and VSLs loss could be reduced to 0.08% and 350 billion CNY (2.3%), respectively."

As for line 32-34, we kept the statement of "The Chinese government should promote

the air pollution control policies that jointly reduce both PM2.5 pollution and ozone pollution" since it is related to our simulation on PM2.5 and ozone concentration change. However, we deleted the statement related to people adjusting their lifestyle according to the air quality information.

4. Section 3.1 should summarize how much NOx and VOC emissions change by scenario, to help interpret some of the ozone concentration results in the coming sections. This information is currently in the Appendix figures, but should be summarized in the main text.

Response: Thank you very much for your suggestion. We summarized NOx and VOC emissions in the manuscript from line 169 to 173 as "NOx emissions are 10 million ton in 2000, and will increase to 32 million in woPol scenario in 2030. However, in 2030 it will reduce to 24 million ton in wPol scenario and 12 million ton in wPol2 scenario. VOC emissions also increase from 16 million in 2000 to 30 million ton in 2030 in woPol scenario. But in wPol and wPol2scenario, it will reduce to 20 million ton and 10 million ton in 2030."

5. Lines 192-202: suggest providing some indication of how many of the provinces with the highest ozone levels fall into each of these three categories. And how many of them might be most affected by the portion of what is called natural, that is actually from international transport of air pollution, wildfires, and farmland. These may not be natural.

Response: We agree that the term "natural" we used in our original manuscript is confusing and not exact. We changed the "natural background" to "zero-out". . In this scenario, all the Chinese emissions from industrial, energy, domestic, traffic, and agricultural (only NH3) sectors are assumed to be zero, and the modeled ozone concentrations are contributed by Chinese natural and foreign (both natural and anthropogenic) sources. This concentration is defined as "zero-out" in our case. We added the number of provinces in each group in the manuscript in line 214-223. "The first group (including

seven provinces) is natural source-dominated provinces where human activity source is lower than 20%, including Xinjiang, Hainan, Qinghai, Gansu, Tianjin, Shanghai and Inner Mongolia. In these provinces that are home to tens of millions of people, daily maximum 8-hour ozone concentration reduction in wPol scenario is not significant, implying that the health damage caused by ozone pollution is hard to mitigate by policy intervention. The second group (including fifteen provinces) is where the human activity source is between 20% to 40%, including Beijing, Hebei, Shanxi, Liaoning, Jilin, Jiangsu, Heilongjiang, Shandong, Henan, Guangdong, Guangxi, Sichuan, Yunnan, Shaanxi and Ningxia. In the third group (eight provinces), anthropogenic emissions dominate (>40%), including Zhejiang, Anhui, Fujian, Jiangxi, Hubei, Hunan, Chongqing and Guizhou".

6. Line 201: "a lot": by how much specifically?

Response: Figure 2 concentration change in 2030 shows the ozone reduction in anthropogenic emissions dominated provinces is about 20-40 $\mu$g/ m$^3$ in different provinces.

7. Figure 3: shift x-axis labels to line up with tick points

Response: We shifted the x-axis label as you suggested.

8. Line 2014: "opportunities to obtain ozone-related diseases, premature death and payment: " is strange wording and should be revised.

Response: We changed "opportunities to obtain ozone-related diseases, premature death and payment" to "risk of ozone-attributable diseases, premature death and health expenditure" (Line 236).

9. Line 235: 61.2% seems off. Is this calculated by dividing 140,100/491,000?

Response: Here we wanted to show these four provinces share the most avoided mortality in China from ozone concentration reduction. The avoided mortality is 36 thousand people in these four provinces and 92 thousand people in China. The ratio is

39%. We modified the sentence Line 256 and 257 in the manuscript as follow: "In wPol scenario, the total reduction of mortality in these four provinces is 36 thousand people, accounting for 39% of national reduction of mortality from ozone pollution".

10. Lines 238 and 242: 4-5% and 1-2% of what? Clarify what these percentages represent.

Response: Percentage 1-2% or 4-5% here is average risk to get ozone-related illness for one person in one year, which is related with ozone concentration and population age. We modified Line 260-265 as "People in these provinces have a higher risk, about 4-5% suffering from health effects such as asthma attacks, respiratory hospital admission, allergic rhinitis, acute respiratory symptoms and coughs from ozone exposure. In contrast, provinces in the east of China with lower daily maximum 8-hour ozone concentration, such as Tianjin, Jiangsu, Beijing and Shandong, are at a lower risk, about 1-2 %, of suffering from such health effects of ozone exposure."

11. Figure 4: How were different morbidity endpoints added together to get the results in the 2nd row? Is this equating all cases of different morbidity endpoints, so that the total number of cases is the cases of asthma + the cases of respiratory hospital admission + cases of cough, etc.?

Response: We wanted to show the risk for one person to get ozone-related diseases in one year in 30 provinces. We added different diseases or health endpoints (Table A1) together and divided the total population in each province.

12. Line 297: why does VSL change by scenario?

Response: Thank you very much for your comments. There maybe are some confusing description about the value of statistical life(VSL). VSL doesn't change by scenario. But the mortality changes in different scenarios, because the ozone concentration is different. The total monetized values of life are different in two scenarios. We have changed the terminology in our manuscript.

13. Line 342: Can forest burning and farmland really be considered natural?

Response: Emissions from forest burning and farmland cannot be considered natural emission. But the impact from air pollution control policy on forest burning and farmland emissions is very limited. For policy assessment, usually we focus on the anthropogenic emissions from industrial, energy, domestic, traffic, and agricultural sectors. To avoid confusion, we called this scenario as "zero-out" instead of "natural" sources.

14. Line 352: except in the cases where you found ozone increases:

Response: Reduction of primary air pollution emissions will lead to ozone decreasing in China. However, in some regions, such as Beijing, Shanghai and Guangdong, under intensive control policy, ozone concentration increase (Figure 2).

15. Line 356-357: There are health effects below the WHO guideline. There is a large body of epidemiological literature on this. For example, see EPA PM National Ambient Air Quality Standards Regulatory Impact Assessments. Suggest removing.

Response: Thank you very much for your suggestion. We removed this sentence in our manuscript.

16. Line 362: Odd to introduce PM mortality results for the first time in the Discussion. Perhaps this should be woven into the health results section or left out completely since no methods information has been given to describe what was done to calculate that (e.g. shape of concentration-response function, epidemiological studies used, age ranges included)

Response: As stated in our response to 1a, this study improved the concentration simulation of PM2.5 with GEOS-Chem and updated a key parameter in the health model, therefore, the results of PM2.5 impacts is not a simple repetition of our paper in 2016. So we adopt your suggestion to move it in results part as a separate section 3.4 "compare PM2.5 and ozone impacts". Moreover, we modified the whole storyline of this manuscript by adding an additional research question of comparing PM2.5 and

ozone, and adding this comparison in the abstract.

17. Line 363-364: what about cardiovascular impacts?

Response: Many studies show ozone pollution increases the cardiovascular disease and mortality. However, from the viewpoint of risk rate, the risk on upper respiratory symptoms and asthma is much higher than cardiovascular impact(Turner, Jerrett et al. 2016).

18. Line 366: What does "2.3 times per capita per year" mean? Does this mean 2.3 cases per capita per year?

Response: Here the 2.3 times should be 2.3%. We made a mistake. Thank you for your consideration.

19. Line 376: "On the other hand" seems unnecessary since this line agrees with the last.

Response: We deleted the "On the other hand" in our manuscript as you suggested.

20. Figure 6: What is changing in the inputs underlying the results in this figure? Just concentrations? What about population, disease rates, etc.? Please clarify.

Response: In our study, the population for different scenarios and air pollutants is same. The disease rate from one pollution in different year is also same. The variables in our study is the PM2.5 and ozone concentration depending on different scenarios.

21. Figure 6: Clarify that the color legend at the bottom applies only to the bottom panel – e.g. by drawing a darker line between the 3rd and 4th rows. I was scratching my head about why there was so much asthma mortality in the top panels, because asthma is gray in the bottom panel.

Response: The color legend at the bottom applies only to the bottom panel for the per capita morbidity. To avoid confusing, we made a new figure with four different panels.

[Figure]

22. Section 6.1: see major comment on health impact assessment above.

Response: We improved our method part and please find it in the support information.

23. Lines 428-442 are perplexing. Many of these variables are not in the equations.

Response: We checked the equations carefully and deleted the variables which are not in the equations for this study. Please see the health assessment model in Support Information we have modified.

24. Line 426: Why is Ir all cause? Is this because only all-cause mortality for ozone exposure was calculated? If cause-specific mortality is calculated, the baseline disease rate should be cause-specific too.

Response: For long-term exposure to ozone, we calculated the all-cause mortality, because we only want to show the total mortality from ozone pollution. But for PM2.5, the non-linear ERFs include cause-specific RR, we still need the cause-specific information in the health assessment model.

25. Lines 445-449: So is this work lost due to death, or morbidity? This whole section is vague and confusing and should be revised.

Response: The work time loss includes two parts, work day loss from morbidity and labor loss from mortality (age 15-65 years). We used the ERF from Extern E for work day loss and minor activity restricted day. For the work loss from premature death, we calculated total mortality and adjusted by age 15 to 65.

26. Line 448: Only 4% of chronic mortality is aged between 15-65 years? That seems really low. Can provide some data to back this up? Need a source.

Response: For ozone pollution, we cannot calculate age-specific mortality because there is no age-specific exposure-response function. We calculated the total mortality from ozone exposure and adjusted by the historical age-specific mortality data from both Chinese Health Statistics and Global Burden of Disease Study. We used the total

mortality by 5-year age group from Chinese Health Statistics from 2002 to 2013 and estimated the mortality for population from 30 to 65 years old, which is around 4% of total mortality. We also checked the age-specific all-cause mortality for the Global Burden of Disease Study 2013. Based on age-specific all-cause mortality from Global Burden of Disease Study 2010, they provided age-specific mortality in 1979, 1990 and 2010. The mortality from 30-64 years old is about 25%, 26% and 29% of total all-cause mortality, respectively. We assume 27% of total chronic mortality is aged between 30 and 65 years old, which is average assumption of the Global Burden of Disease Study 2010(Wang, Dwyer-Lindgren et al. 2012). We modified the sentence Line 50-52 in Support Information as "Based on age-specific all-cause mortality from Global Burden of Disease Study 2010, We assume 27 of total chronic mortality is aged between 30 and 65 years old, which is average assumption of the Global Burden of Disease Study 2010."

27. Line 784 is repetitive

Response: We deleted the repetitive sentence as you suggested.

28. Line 788: missing section reference

Response: We modified Line 400 to 402 in Support Information as "Besides the anthropogenic emissions of ozone precursors, we also consider the contribution of natural sources."

Reference

Apte, J. S., J. D. Marshall, A. J. Cohen and M. Brauer (2015). "Addressing global mortality from ambient PM2. 5." Environmental science & technology 49(13): 8057-8066. Berman, J. D., N. Fann, J. W. Hollingsworth, K. E. Pinkerton, W. N. Rom, A. M. Szema, P. N. Breysse, R. H. White and F. C. Curriero (2012). "Health benefits from large-scale ozone reduction in the United States." Environmental health perspectives 120(10): 1404-1410. Bickel, P., R. Friedrich, B. Droste-Franke, T. Bachmann, A. Gressmann,

A. Rabl, A. Hunt, A. Markandya, R. Tol and F. Hurley (2005). ExternE Externalities of Energy Methodology 2005 Update. Hu, L., D. J. Jacob, X. Liu, Y. Zhang, L. Zhang, P. S. Kim, M. P. Sulprizio and R. M. Yantosca (2017). "Global budget of tropospheric ozone: evaluating recent model advances with satellite (OMI), aircraft (IAGOS), and ozonesonde observations." Atmospheric Environment 167: 323-334. Jeong, J. I. and R. J. Park (2013). "Effects of the meteorological variability on regional air quality in East Asia." Atmospheric environment 69: 46-55. Matus, K., K.-M. Nam, N. E. Selin, L. N. Lamsal, J. M. Reilly and S. Paltsev (2012). "Health damages from air pollution in China." Global Environmental Change 22(1): 55-66. Mu, Q. and H. Liao (2014). "Simulation of the interannual variations of aerosols in China: role of variations in meteorological parameters." Atmospheric Chemistry and Physics 14(18): 9597-9612. Park, R. J., M. J. Kim, J. I. Jeong, D. Youn and S. Kim (2010). "A contribution of brown carbon aerosol to the aerosol light absorption and its radiative forcing in East Asia." Atmospheric Environment 44(11): 1414-1421. Pope III, C. A., R. T. Burnett, M. J. Thun, E. E. Calle, D. Krewski, K. Ito and G. D. Thurston (2002). "Lung cancer, cardiopulmonary mortality, and long-term exposure to fine particulate air pollution." Jama 287(9): 1132-1141. Samir, K. and W. Lutz (2014). "The human core of the shared socioeconomic pathways: Population scenarios by age, sex and level of education for all countries to 2100." Global Environmental Change. Selin, N. E., S. Wu, K.-M. Nam, J. M. Reilly, S. Paltsev, R. G. Prinn and M. D. Webster (2009). "Global health and economic impacts of future ozone pollution." Environmental Research Letters 4(4): 044014. Trivitayanurak, W., P. Palmer, M. Barkley, N. Robinson, H. Coe and D. Oram (2012). "The composition and variability of atmospheric aerosol over Southeast Asia during 2008." Atmospheric Chemistry and Physics 12(2): 1083-1100. Turner, M. C., M. Jerrett, C. A. Pope III, D. Krewski, S. M. Gapstur, W. R. Diver, B. S. Beckerman, J. D. Marshall, J. Su and D. L. Crouse (2016). "Long-term ozone exposure and mortality in a large prospective study." American journal of respiratory and critical care medicine 193(10): 1134-1142. Viscusi, W. K. and J. E. Aldy (2003). "The value of a statistical life: a critical review of market estimates throughout the

world." Journal of risk and uncertainty 27(1): 5-76. Wang, H., L. Dwyer-Lindgren, K. T. Lofgren, J. K. Rajaratnam, J. R. Marcus, A. Levin-Rector, C. E. Levitz, A. D. Lopez and C. J. Murray (2012). "Age-specific and sex-specific mortality in 187 countries, 1970–2010: a systematic analysis for the Global Burden of Disease Study 2010." The Lancet 380(9859): 2071-2094. Wang, H. G., M. Montoliu-Munoz and N. F. D. Gueye (2009). "Preparing to Manage Natural Hazards and Climate Change Risks in Dakar, Senegal: A Spatial and Institutional Approach." Wang, Y., Y. Zhang, J. Hao and M. Luo (2011). "Seasonal and spatial variability of surface ozone over China: contributions from background and domestic pollution." Atmospheric Chemistry and Physics 11(7): 3511-3525. West, J. J., S. J. Smith, R. A. Silva, V. Naik, Y. Zhang, Z. Adelman, M. M. Fry, S. Anenberg, L. W. Horowitz and J.-F. Lamarque (2013). "Co-benefits of mitigating global greenhouse gas emissions for future air quality and human health." Nature climate change 3(10): 885-889. Xie, X. (2011). "The value of health: Applications of choice experiment approach and urban air pollution control strategy. Ph.D. thesis. Peking University." Xie, Y., H. Dai, H. Dong, T. Hanaoka and T. Masui (2016). "Economic impacts from PM2. 5 pollution-related health effects in China: A provincial-level analysis." Environmental Science & Technology 50(9): 4836-4843. Yan, Y., J. Lin, J. Chen and L. Hu (2016). "Improved simulation of tropospheric ozone by a global-multi-regional two-way coupling model system." Atmospheric Chemistry and Physics 16(4): 2381-2400.

Please also note the supplement to this comment:
https://www.atmos-chem-phys-discuss.net/acp-2017-849/acp-2017-849-AC1-supplement.zip